# THE COST OF DELEGATION

## ABSTRACT

We study reinforcement learning and alignment through the lens of hierarchical coordination, where a principal steers many delegates with partial views and coupled effects. Starting from nonlinear dynamics, we identify the Cost of Delegation as the performance gap between centralized and decentralized control, decomposed into delegation, coordination, information, and surrogate mismatch components. We bound CoD, show that information value is decision-theoretic, and discuss implications for modern systems. Our work provides a theoretical foundation and new perspective for designing robust, scalable multi-agent systems.

## 1 INTRODUCTION

> *The sailors are quarrelling with one another about the steering... But that the true pilot must pay attention to the year, seasons, sky, stars, and winds.*
>
> > *Plato*

> *We are like sailors who must rebuild their ship on the open sea, never able to dismantle it in dry dock and reconstruct it from the best materials.*
>
> > *Otto Neurath*

We study a foundational problem in reinforcement learning: how to understand and characterize the structural cost of alignment. Under the current paradigm, alignment is often formulated as an optimization problem. Given a reward function or its learned surrogate, the goal is to find a policy that maximizes the expected cumulative reward. RLHF (Ouyang et al., 2022), GRPO (Shao et al., 2024), Constitutional AI (Bai et al., 2022), DPO (Rafailov et al., 2023)) and other pipelines or variants follow this logic. Such methodology implicitly assumes the existence of a goal that is attainable in principle, and the task of the algorithm is to approximate it efficiently and robustly. Alignment is thus broadly understood and framed as an optimization task with engineering challenges.

However, the scale of modern systems makes direct control infeasible in practice. While we optimize parameters via gradients, we lack direct, interpretative control over the internal state dynamics that instantiate alignment. As a consequence, influence is exerted through intermediate mechanisms. For example, reward models typically project human preferences as scalar signals, constitutional principles decompose the alignment goal into verifiable principles, and preference data indirectly shape behavior through gradients. These mechanisms constitute a hierarchical structure (whether explicit or implicit), where high-level intentions must be interpreted and executed by numerous subsystems or modules. A natural question arises: since we are forced to achieve alignment through a hierarchical and modular architecture, will different architectural choices lead to diverse alignment costs? Are there structural patterns that transcend specific pipelines?

This is our starting point. We now ask whether there exists a component of the alignment cost that is neither a product of algorithmic flaws nor a result of insufficient data, but rather a structural cost that persists even under perfect optimization and perfectly "well-intentioned" conditions. Once adopting the reality of hierarchical coordination, the core issue naturally shifts from *how* to design better reward functions to *what* is the irreducible cost of delegation itself? *How* does it decompose into tractable components? *Which* parts can be controlled through architectural design? We name such structural gap the **Cost of Delegation** (CoD).

**Our contributions.** We formalize the framework as multi-component systems with partial observation and coupling effects. In some cases, this manifests as an explicit hierarchical structure, such as in RLHF or multi-agent systems. Regardless of the specific architecture, the crux is how coordination and information structures affect overall performance. We analyze these dynamics through a Linear-Quadratic (LQ) surrogate under certainty equivalence. This method has been proved to be statistically effective for LQ control in modern RL theory (Mania et al., 2019). Crucially, current post-training pipelines generally relies on a certainty-equivalence-like logic. This choice allows us to derive closed-form bounds for structural costs, providing an analyzable proxy for understanding the local curvature of the alignment landscape in nonlinear systems. We then establish a four-layer policy hierarchy from centralized optimality to realistic delegation. We prove that the gaps between each layer are non-negative and identify which gaps can be explicitly bounded. This induces a telescoping decomposition of the LQ structural gap between centralized and delegated performance into an information term, a coordination-structure term, and a residual delegation term.

**Implications.** We find that in our framework, the relevant notion of information value is decision-theoretic in nature: what matters is how observations change optimal actions, not how much entropy they carry. Specifically, the value derived from observing a particular direction depends on the sensitivity of that direction to control decisions rather than the statistical variance of that direction. This means that high-variance but decision-independent directions (such as those favored by PCA) may offer no benefit while low-variance but decision-sensitive ones may be crucial. Our experiments support this prediction. In the content moderation task, variance-based observation performs comparably to random projection, while observation aligned with the decision boundary significantly reduces delegation costs. Thus, reward models should focus on characterizing those preference distinctions that have substantial impact on policy behavior, rather than trying to capture all preference variations uniformly. Similarly, routing in MoE may benefit from assigning experts based on gradient sensitivity (task relevance) rather than input feature clustering (statistical characteristics).

Recent research supports our insight. Chen et al. (2024) show that the response length in the reward model is a high-variance but decision-irrelevant signal, and decoupling it from the quality signal can significantly alleviate reward hacking. DeepSeekMoE (Dai et al., 2024) achieves expert specialization through fine-grained expert segmentation to reduce the overlap of redundant knowledge. In short, we offer a new perspective on understanding intrinsic mechanisms of alignment and designing robust, scalable modern systems.

**Roadmap.** Section 2 introduces the general principle and four cost sources. Section 3 & 4 establish a formal framework, including hierarchical coordination model, four-level policy hierarchy, and telescoping decomposition. Section 5 discusses explicit bounds for different structural components and the global bound. Section 6 provides evidence for the core predictions based on decision-weighted information. Section 7 discusses the implications for modern systems.

## 2 TOP-DOWN ANATOMY

### 2.1 FIRST PRINCIPLE

We start with a first principle: In any system that achieves its goal through intermediate mechanisms, the introduction of constraints inevitably leads to a performance penalty. Let $J(\pi; M)$ denote the reward of policy $\pi$ in environment $M$, with higher values being better[1].

**Definition 2.1.** *Given two policy classes, where $\Pi_{rich}$ is less restricted and $\Pi_{constrained}$ is more restricted, and $\Pi_{rich} \supset \Pi_{constrained}$, the **Cost of Delegation** (under given perspective) is defined as:*

$$CoD_{Component} = \max_{\pi \in \Pi_{rich}} J(\pi; M) - \max_{\pi \in \Pi_{constrained}} J(\pi; M).$$

---

[1]**Remark on notations:** Fundamentally $J$ is the objective; in RL it is interpreted as reward. Following the conventions of control theory, under later LQ specification, $J$ represents quadratic cost. This sign change does not affect any substantial conclusion: a reduction in return is equivalent to an increase in cost.

Non-negativity is directly guaranteed by the set inclusion relationship. Optimization over a larger feasible region cannot yield a worse solution. CoD characterizes the reward loss (gap) relative to the unconstrained optimum that is inherently brought about by the constraints themselves. Limitations on information acquisition, component coordination, model fidelity, and optimization capabilities can all be incorporated into this framework.

## 2.2 FOUR SOURCES

More specifically, following a source-of-constraint logic, CoD can be stratified along four axes.

**(A) Surrogate mismatch.** The alignment system optimizes on the surrogate objective rather than directly on the true objective. Let

$$\pi^*_{\text{true}} = \arg\max_\pi J(\pi; M_{\text{true}}), \quad \pi^*_{\text{surrogate}} = \arg\max_\pi J(\pi; M_{\text{surrogate}}).$$

The optimal strategies under the real target and the surrogate target respectively. The cost brought by surrogate mismatch is

$$\text{CoD}_A = J(\pi^*_{\text{true}}; M_{\text{true}}) - J(\pi^*_{\text{surrogate}}; M_{\text{true}}).$$

This characterizes "how much return is lost in the *real* environment relative to the *optimal* one due to using an approximate objective for optimization". Conceptually, optimizing a surrogate imposes an implicit constraint. It restricts the effective solution to the set of policies favored by the surrogate gradients, rather than the true ones. Non-negativity is guaranteed by the optimality of $\pi^*_{\text{true}}$ on $M_{\text{true}}$. Recent research supports this perspective (Zhuang & Hadfield-Menell, 2020).

**(B) Information constraints.** Delegates can only observe a portion of the state. Let $\Pi_{\text{full}}$ be a policy class based on full observation, and $\Pi_{\text{partial}}$ be a policy class based on partial observation. Any policy that depends only on partial information can be implemented under full information, therefore $\Pi_{\text{full}} \supset \Pi_{\text{partial}}$. The cost of information constraints is:

$$\text{CoD}_B = \max_{\pi \in \Pi_{\text{full}}} J(\pi; M) - \max_{\pi \in \Pi_{\text{partial}}} J(\pi; M).$$

This formalization is conceptually consistent with Blackwell (1953)'s classic result on the value of information that a finer information structure supports better decision-making.

**(C) Coordination constraints.** When a system consists of multiple delegates, dense global coordination is impractical. The system must rely on sparse local interactions, where only adjacent delegates can coordinate. Let $\Pi_{\text{dense}}, \Pi_{\text{sparse}}$ be the policy class that can be implemented under dense or sparse coordination respectively. It is generally considered that dense structures can simulate any sparse strategy, therefore $\Pi_{\text{dense}} \supset \Pi_{\text{sparse}}$. The cost of the coordination constraint is:

$$\text{CoD}_C = \max_{\pi \in \Pi_{\text{dense}}} J(\pi; M) - \max_{\pi \in \Pi_{\text{sparse}}} J(\pi; M).$$

This reflects the core problem in team decision theory (Radner, 1962; Marschak & Radner, 1958).

**(D) Training residual.** Even given the architecture and objective function, the training algorithm may not find the optimal policy under that setting. Let $\pi^* = \arg\max_{\pi \in \Pi} J(\pi; M)$ be the optimal policy in the policy class, and $\hat{\pi} \in \Pi$ be the actual trained policy. The training residual is:

$$\text{CoD}_D = J(\pi^*; M) - J(\hat{\pi}; M).$$

This reflects both computational and statistical constraints. The realizable policy set is restricted by the algorithm's convergence properties and the available training data. Nonnegativity is directly guaranteed by the definition of optimality of $\pi^*$.

## 2.3 BRIDGE (GAP) TO REALITY

Section 2.1 and 2.2 define CoD from first principles. Now back to real-world systems. Given a *fixed* objective function (A). The ideal benchmark is a single optimizer with complete information. Real-world systems, however, involve multiple delegates operating under partial observations and

sparse coordination. Thus following this performance loss path, the structural component of CoD can be decomposed into three steps.

**Step 1. Delegation itself ($\Delta_{\mathbf{deleg}}$).** Even when all delegates share the same objective, replacing a single optimizer with a multi-agent game can incur an arbitrarily large performance penalty. Witsenhausen (1968)'s counterexample shows that, even under linear dynamics, quadratic costs, and perfect cooperation, the optimal decentralized policy can be a highly nonlinear, computationally intractable signaling rule. This impossibility result delineates the scope of our analysis.

**Step 2. Sparse coordination ($\Delta_{\mathbf{coord}}$).** Given that delegates are making their own decisions, dense coordination (where each delegate interacts with all other delegates) is not feasible in practice. The system must degenerate to sparse coordination (interacting only with local neighbors). This step corresponds to (C), and its gap depends on the topology of the coordination graph.

**Step 3. Partiality of observation ($\Delta_{\mathbf{info}}$).** Given the coordination structure, delegates are also limited to observing only a portion of the state. This step corresponds to (B), and its gap depends on the design of the observation structure $\Pi$.

The gaps in the second and third steps are bounded. They depend on the specific architecture choices $W$ and $\Pi$, which is exactly what we will discuss in the following sections.

## 3 PRELIMINARY

### 3.1 LQ AND CE

We use Linear-Quadratic (LQ) surrogate combined with the certainty equivalence (CE) principle as the proxy model. Certainty equivalence is a two-stage method: first, system parameters are estimated based on observed data; then, the estimated values are treated as true values for optimal control design. Our choice is chosen based on three considerations.

**1.** LQ framework is a natural approximation of a nonlinear system near its operating point. For dynamics $f$ and cost function satisfying appropriate smoothness conditions, the first two orders of the Taylor expansion yield linear dynamics and a quadratic cost structure. When the system trajectory is concentrated around a nominal state $\bar{\phi}$, higher-order remainder terms are controllable. This perspective is well established in stochastic control theory (Anderson & Moore, 2007).

**2.** CE has been shown to possess statistical validity in modern reinforcement learning theory. A series of works demonstrate that for LQ control problems, certainty-equivalent controllers based on finite-sample estimates achieve optimal rates of convergence (Mania et al., 2019; Dean & Recht, 2021), and that policy gradient methods enjoy global convergence guarantees in the LQ setting (Fazel et al., 2018; Cohen et al., 2019). These results show that LQ surrogates are not only mathematically tractable but also statistically efficient.

**3.** CE shares structural similarity with post-training pipelines. In RLHF, the learned reward model is treated as a nominal objective function for subsequent policy optimization (Rafailov et al., 2023). In MoE, the router makes deterministic expert assignments based on current representations ((Fedus et al., 2021)). In multi-agent orchestration, the orchestrator allocates tasks based on state estimates. CE provides a unified analytical perspective for understanding modern systems.

### 3.2 ASSUMPTIONS

The system relies on the following two classes of assumptions. Formal statements see Appendix A.

**System regularity assumptions (S1–S4).** S1 assumes that the dynamics $f : \mathcal{S} \times \mathbb{R}^d \to \mathcal{S}$ is Lipschitz continuous and twice differentiable on the operating domain, the process noise is a sub-Gaussian martingale-difference sequence, the state space $\mathcal{S}$ is compact, and all action sets are bounded. These conditions justify the local LQ approximation and control the Taylor remainder.

**S2** specifies that the principal observes three noisy channels: state, response, and reward. **S3** specifies that each delegate $m$ observes a local state $\phi_{m,t} = \Pi_m \phi_t + \nu_{m,t}$, where the projection family $\{\Pi_m\}$ has bounded operator norm $\|\Pi_m\|_2 \leq 1$ and satisfies a coverage condition. **S4** imposes a persistent-excitation condition on the predictable regressors, ensuring that the reduced-form model can be identified from finite samples with standard concentration guarantees.

**Game regularity assumptions (G1–G2).** **G1** contains three subconditions: (a) $\lambda_{\min}((G + G^T)/2) \geq m > 0$ implies strong monotonicity of the game gradient and therefore existence and uniqueness of the Nash equilibrium; (b) the coordination matrix is bounded, $\|W\|_2 \leq w_{\max}$; (c) $W$ admits one of three tractable structures (Low-rank + sparse; Tree/DAG; Block-sparse), which reduces equilibrium computation from the naive $O((Md)^3)$ to near-linear complexity in the number of agents. **G2** requires the closed-loop Jacobian to satisfy $\sup_{(\phi, u_P) \in \mathcal{S} \times \mathcal{U}_P} \|D_\phi F(\phi; u_P)\|_2 < 1/\gamma$ which ensures that the discounted accumulation of surrogate errors is bounded.

## 4 FRAMEWORK

Recall how the three-step decomposition in Section 2.3 is mapped onto the ABCD axes in Section 2.2. The three-step decomposition assumes given objective function and system parameters, so (A) and (D) are orthogonal to these three steps. After Section 3 adopts the LQ-CE framework, (A) enters the framework through the surrogate approximation error $A(\delta\phi)$. (D) will be handled via the distinction between epistemic and persistent errors in Section 5.

Table 1: Mapping.

| Four Sources | Three Steps | Bound |
|---|---|---|
| (A) | – | Expr. 5.3 |
| (B) | $\Delta_{\text{info}}$ | Expr. 5.2 |
| (C) | $\Delta_{\text{coord}}$ | Expr. 5.1 |
| (D) | – | Expr. 5.6 |
| – | $\Delta_{\text{deleg}}$ | unbounded |

Step 1 ($\Delta_{\text{deleg}}$) does not fall within the ABCD taxonomy because it is not a structure-design problem. Witsenhausen (1968)'s counterexample proves that such gap can be arbitrarily large, so the objective is to minimize it through mechanism design, which is a seperate question. At the same time, real systems rarely implement such such precise design and far from perfect alignment between local/global objectives. Thus, we treat $\Delta_{\text{deleg}}$ as a given parameter rather than something we can structurally bound.

### 4.1 EMERGENT COORDINATION GAME

We need a mathematical structure to model the interaction of multiple delegates. We start from the delegates' local objectives and derives how the coordination matrix $W$ emerges naturally from the system structure, and then the resulting equilibrium. This formalization serves two purposes: first, to show that $W$ is not exogenously designed; second, to provide the foundation for the four-layer hierarchy defined later, where each layer corresponds to a different configuration of $(W, \Pi)$, and the equilibrium characterization determines the optimal value $J^*$ at each layer.

Consider a system with a principal and $M$ delegates. The state $\phi_t \in \mathcal{S} \subseteq \mathbb{R}^n$ evolves according to $\phi_{t+1} = f(\phi_t, a_t) + w_t$, where $a_t = \sum_m u_{m,t}$ is the aggregate action of the delegates. The principal holds a global objective but cannot directly control the system. Instead, each delegate $m$ chooses actions based on its local observation $\phi_{m,t} = \Pi_m \phi_t$ so as to minimize the local objective

$$\mathbb{E}\left[ \sum_{t=0}^{\infty} \gamma^t \left( \frac{1}{2} \|\Pi_m \phi_t - \psi_m(u_{P,t})\|_{Q_m}^2 + \frac{1}{2} \|u_m\|_{R_m}^2 \right) \right],$$

where $\psi_m(u_P)$ denotes the target state induced by the principal's command. We can clearly see that delegate $m$'s action choice depends on the actions taken by other delegates, because all actions jointly influence future states through the shared dynamics $f$. This strategic interdependence forms the foundation of the game-theoretic structure. We next derive from first principles how delegation games emerge from objective functions.

**Theorem 4.1.** *(Emergent Coordination) Under certainty equivalence and local linearization, the infinite-horizon problems reduce (to second order) to the single-stage game*

$$J_m(u_m; u_{-m}) = \frac{1}{2} u_m^T R_m u_m + u_m^T K_m + \frac{1}{2} \sum_{j=1}^M u_m^T W_{mj} u_j + \varepsilon_m,$$

*where $K_m = (\Pi_m B)^T Q_m [(\Pi_m A)\delta\phi - \psi_m(u_P)]$, and the coordination matrix is given by $W_{mj} = (\Pi_m B)^T Q_m (\Pi_j B)$. The approximation error satisfies*

$$|\varepsilon_m| \leq C_h^{(m)} \|\delta\phi\|_2^3 + C_f^{(m)} (\|\delta\phi\|_2^2 + \|a\|_2^2).$$

The coordination matrix emerges mechanically from the objective function: the action of delegate $j$ is propagated to the state space through the input matrix $B$, and then weighted by delegate $m$'s cost matrix $Q_m$ and projection $\Pi_m$ to generate coupling. Whenever $(\Pi_m B)^T Q_m (\Pi_j B) \neq 0$, there is a coordination requirement between the two delegates. This structure induces a graph $\mathcal{G} = (V, E)$, where the vertex set is $V = \{1, \ldots, M\}$ and there is an edge $(m, j) \in E$ if and only if $W_{mj} \neq 0$. The quadratic–cubic form of $\varepsilon_m$ is the natural consequence of truncating the Taylor expansion at the lowest tractable order, see Section 5.2 for details.

**Proposition 4.1.** *(Nash Equilibrium) Let $G := R + W$, the aggregate game*

$$J(\mathbf{u}) = \frac{1}{2} \mathbf{u}^T G \mathbf{u} + \mathbf{u}^T K(\delta\phi, u_P)$$

*admits a unique Nash equilibrium $\mathbf{u}^* = -G^{-1}K$ under Assumption G1(a), where $\delta\phi := \bar{\phi} - \phi$ and the equilibrium mapping is Lipschitz in $(\delta\phi, u_P)$.*

When $W$ is symmetric, it is a potential game (Monderer & Shapley, 1996) with potential $\Phi(\mathbf{u})$. So the PNE is simply the minimizer of $\Phi$, implying $\Delta_{\text{deleg}} = 0$, and whether satisfied is, as discussed, a mechanism design objective orthogonal to our focus: one must design the delegates' objective functions $\{J_m\}$ so that the induced $W$ is symmetric. This motivates future works.

### 4.2 FOUR-LEVEL POLICY HIERARCHY AND TELESCOPING DECOMPOSITION

Proposition 4.1 characterizes the equilibrium for a given triplet $(R, W, K)$. In practice, however, real systems face two additional constraints: the coordination matrix may be sparsified due to computational limitations, and observations may be made partial due to architectural constraints. To measure the performance impact of these constraints, we define the following four-level policy hierarchy.

| Level | Decision | Info | Coord. |
|-------|----------|------|--------|
| L1 | Centralized opt. | Full | — |
| L2 | Nash eq. | Full | Dense $W^*$ |
| L3 | Nash eq. | Full | Sparse $W_k$ |
| L4 | Nash eq. | Partial | Sparse $W_k$ |

L1 is the ideal benchmark: a single optimizer directly minimizes the global objective. L2 introduces delegation while retaining full information and full coordination. L3 sparsifies the coordination matrix from $W^*$ to $W_k$. L4 further restricts observations from full to partial.

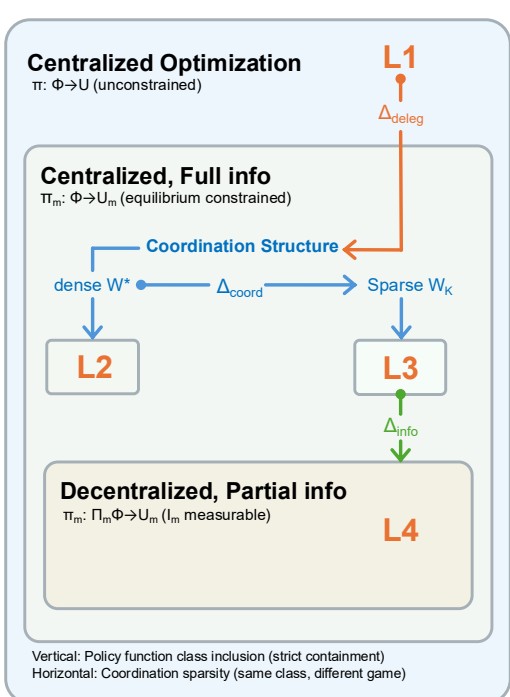

Figure 1: Policy-space architecture.

**Theorem 4.2.** *(Monotonicity) Under G1(a), let $J^*_{L\ell}$ be the optimal value at Levels L1–L4. Then*

$$J^*_{L1} \ \le \ J^*_{L2} \ \le \ J^*_{L3} \ \le \ J^*_{L4},$$

*and when $W$ is symmetric, $J^*_{L1} = J^*_{L2}$.*

The four-level hierarchy is designed so that each adjacent transition changes exactly one structural factor. The gaps between adjacent levels thus isolate the cost contribution of each single factor.

**Proposition 4.2.** *(Structural Gaps) The three-step decomposition is quantified by the gaps between adjacent levels, and by monotonicity, each term is nonnegative.*

$$\Delta_{deleg} := J^*_{L2} - J^*_{L1}, \qquad \Delta_{coord} := J^*_{L3} - J^*_{L2}, \qquad \Delta_{info} := J^*_{L4} - J^*_{L3}.$$

**Expression 4.1.** *(Telescoping Decomposition) Within the LQ surrogate, CoD satisfies:*

$$CoD_{LQ} = \Delta^{LQ}_{struct} := J^*_{L4} - J^*_{L1} = \Delta_{deleg} + \Delta_{coord} + \Delta_{info}$$

Each term is now explicit. $\Delta_{\text{deleg}}$: centralized optimization to PNE ($L1 \to L2$). $\Delta_{\text{coord}}$: dense to sparse coordination ($L2 \to L3$). $\Delta_{\text{info}}$: full to partial observation ($L3 \to L4$). Together, they form CoD within LQ surrogate, while *full* CoD includes surrogate mismatch and training residuals.

### 4.3 TOY MODEL

To expose the mechanics of the framework, consider two delegates representing helpfulness (H) and safety (S), respectively. The output logit is their controlled aggregate:

$$o = c_H u_H - c_S u_S, \qquad c_H, c_S > 0.$$

Conflict of objectives: helpfulness pushes $o$ upward, while safety pushes $o$ downward. Given a query $q$, delegate $m$ tracks its target $t_m(q)$ and incurs quadratic tracking plus effort cost:

$$\ell_m(u_m; q) \ = \ \tfrac{1}{2} q_m \left( o - t_m(q) \right)^2 \ + \ \tfrac{1}{2} r_m u_m^2, \qquad q_m \ge 0, \ r_m > 0.$$

Applying Theorem 4.1, the coordination matrix and driving vector are

$$W = \begin{bmatrix} c_H^2 q_H & -c_H c_S q_H \\ -c_H c_S q_S & c_S^2 q_S \end{bmatrix}, \qquad K(q) = - \begin{bmatrix} c_H q_H \, t_H(q) \\ -c_S q_S \, t_S(q) \end{bmatrix}.$$

The off-diagonals $\propto -c_H c_S q_m$ capture the coordination couplings, one delegate's act changes the other's tracking error through the shared output channel. Even in this minimal system, the three components of $CoD_{LQ}$ arise naturally. When $q_H \ne q_S$, $W$ becomes asymmetric, the game loses its potential structure, $\Delta_{\text{deleg}} > 0$. Ignoring the off-diagonal couplings yields $\Delta_{\text{coord}} \propto c_H^2 c_S^2 (q_H^2 + q_S^2)$. Under partial observation, $\Delta_{\text{info}} \propto \mathrm{Var}(t_m(q) \mid \Pi_m q)$.

## 5 QUANTIFYING THE COST OF DELEGATION

Section 4 defined structural gaps pointwise in $(\phi, u_P)$, here we analyze their discounted and expected forms under stationary state distributions and the randomness of finite-sample learning.

### 5.1 COORDINATION COST

Recall Theorem 4.2 and Proposition 4.2:

$$\Delta_{\text{coord}} := J^*_{L3} - J^*_{L2} = J(u_k) - J(u^*) = \frac{1}{2} \delta u^T G^* \delta u.$$

**Expression 5.1.** *(Coordination cost) Let $E := W^* - W_k$ denote the sparsification error and $S^* := \frac{1}{2}(G^* + (G^*)^T)$. Under Assumption G1(a), there exists a constant $C_{struct} := \frac{\lambda_{\max}(S^*)}{2 \lambda_{\min}(S^*)^2}$ such that*

$$\Delta_{coord} \ \le \ C_{struct} \|G_k^{-1}\|_2^2 \, \|E\|_F^2 \, \|K\|_2^2.$$

**Intuition** Expression 5.1 highlights three levers that control $\Delta_{\text{coord}}$. $\|E\|_F$ is determined by the sparsification scheme, which motivates the tractable structures in Assumption G1(c) (low-rank plus sparse, tree/DAG, block-sparse), each inducing a different pattern of $E$ that can be tuned to minimize $\|E\|_F$ given the system topology. The factor $\|G_k^{-1}\|_2$ captures how ill-conditioned the sparse game is: if $G_k$ is nearly singular, small coordination errors are amplified into large deviations in equilibrium strategies. Finally, $\|K\|_2$ depends on the current state deviation $\delta\phi$ and the principal's command $u_P$, it is an exogenous input rather than a design variable.

## 5.2 INFORMATION COST

Recall that for stochastic LQ surrogate, we study the expected information gap:

$$\Delta_{\text{info}} := \mathbb{E}\big[J_{\text{L4}}^* - J_{\text{L3}}^*\big].$$

**Notation.** Fix the sparse game $G_k = R + W_k$ and its symmetric part $S_k := (G_k + G_k^\top)/2 \succeq 0$, with stage cost $J(u) := \frac{1}{2}u^\top G_k u + u^\top K$. Let $u^*(\phi) = -G_k^{-1}K(\phi)$ denote the full-information equilibrium and $\hat{u}(\phi) = -G_k^{-1}\hat{K}(\phi)$ the partial-information equilibrium, where $\hat{K}_m(\phi) := \mathbb{E}[K_m(\phi) \mid \Pi_m\phi]$. Define the innovation $\epsilon := K - \hat{K}$ and write $\delta u := \hat{u} - u^* = G_k^{-1}\epsilon$. In the LQ surrogate, $\epsilon$ is a linear function of the state deviation $\delta\phi$ with covariance $\Sigma_\phi$, i.e. $\epsilon = L\,\delta\phi$ and $\text{Cov}(\epsilon) = L\Sigma_\phi L^\top$ for some matrix $L$ determined by $(A, B, Q_m, \Pi_m)$.

**Expression 5.2** (Information cost)**.** *Using the $G_k u^* + K = 0$ and symmetrizing,*

$$\Delta_{\text{info}} = \frac{1}{2}\,\mathbb{E}[\delta u^\top S_k \delta u] = \frac{1}{2}\,\text{tr}\big(G_k^{-T}S_k G_k^{-1}\,\text{Cov}(\epsilon)\big) = \frac{1}{2}\,\text{tr}\big(G_k^{-T}S_k G_k^{-1}\,L\Sigma_\phi L^\top\big).$$

**Intuition.** Expression 5.2 shows that information cost is entirely determined by the *innovation covariance* $\text{Cov}(\epsilon)$ as filtered through the structural weight $G_k^{-T}S_k G_k^{-1}$. More informative observation schemes (in the Blackwell sense) shrink the residual operators $(I - P_m)$ and hence $L\Sigma_\phi L^\top$, monotonically reducing $\Delta_{\text{info}}$. The notion of information value here is decision-theoretic rather than purely statistical: each state direction is weighted not just by its variance in $\Sigma_\phi$, but by its sensitivity under $L^\top G_k^{-T}S_k G_k^{-1}L$. This perspective is somewhat counterintuitive from a purely statistical viewpoint: directions that dominate PCA or clustering criteria may be essentially irrelevant for decision-making, while low-variance but decision-sensitive directions can be crucial.

## 5.3 SURROGATE APPROXIMATION COST

Axis (A) does not contribute to $CoD_{\text{LQ}}$, it enters only as an entry cost $A(\delta\phi_0)$ as the alignment systems optimizes on the surrogate objective rather than true one. Let $\text{err}_t$ be the per-stage mismatch under the same closed-loop policy and define $A(\delta\phi_0; G_k) := \sum_{t=0}^\infty \gamma^t \text{err}_t$.

**Expression 5.3** (Surrogate bound and $A \otimes C$ coupling)**.** *Under S1–S4 and G2 there exist constants $C_2, C_3 > 0$ depending only on local derivatives of $f$ and $h_m$ such that*

$$|\text{err}_t| \le C_3\|\delta\phi_t\|_2^3 + C_2\big(1 + ML_K^2\|G_k^{-1}\|_2^2\big)\|\delta\phi_t\|_2^2.$$

*Let $L_{\text{cl}} = \sup_\phi \|D_\phi F(\phi; u_P)\|_2 < 1/\gamma$. Then*

$$A(\delta\phi_0; G_k) \le \frac{C_2\big(1 + ML_K^2\|G_k^{-1}\|_2^2\big)}{1 - \gamma L_{\text{cl}}^2}\|\delta\phi_0\|_2^2 + \frac{C_3}{1 - \gamma L_{\text{cl}}^3}\|\delta\phi_0\|_2^3.$$

*Define the approximation constants $A = (C_2, C_3)$ and the coordination–stability multipliers $C(G_k) = \left(\frac{1 + ML_K^2\|G_k^{-1}\|_2^2}{1 - \gamma L_{\text{cl}}^2}, \frac{1}{1 - \gamma L_{\text{cl}}^3}\right)$. Then*

$$A(\delta\phi_0; G_k) \le \big(A \otimes C(G_k)\big) \cdot \big(\|\delta\phi_0\|_2^2, \|\delta\phi_0\|_2^3\big).$$

**Intuition.** The quadratic–cubic form in Expression 5.3 comes directly from truncating the Taylor expansions of $h_m$ and $f$ at the lowest order that still admits tractable trajectory-level bounds. It is the best we can compute while keeping the analysis finite dimensional. $A \otimes C$ coupling is now

explicit: the surrogate mismatch (A) is set by local approximation constants $A$, but its magnitude is multiplied by the structural choice of $W_k$ through $G_k^{-1}$ and the closed loop Lipschitz constant $L_{\text{cl}}$.

## 5.4 PERSISTENT VS. EPISTEMIC: THE TOTAL COST OF DELEGATION

Time to connect the dots. We separate the total CoD into a *persistent* component driven by the control problem (architecture and surrogate) and an *epistemic* component driven by learning. $T$ denotes the number of training samples or updates used to fit reduced-form models or policies.

**Expression 5.4** (Total CoD at scale $T$). *Let $J_{\text{ideal}}^*$ be the value of an ideal full-information controller under true dynamics, and $\hat{\pi}_T$ the learned policy after $T$ samples.*

$$\text{CoD}_{tot}(T) := \mathbb{E}\big[J(\hat{\pi}_T; M_{true})\big] - J_{ideal}^* = \text{CoD}_{LQ} + A(\delta\phi) + \text{CoD}_D(T).$$

**Expression 5.5** (Persistent part and structural bound).

$$\text{CoD}_{persistent} := \text{CoD}_{LQ} + A(\delta\phi) = \Delta_{deleg} + \Delta_{coord} + \Delta_{info} + A(\delta\phi).$$

*Collecting the bounds yields an explicit constant $B_{struct}$ such that $\text{CoD}_{persistent} - \Delta_{deleg} \leq B_{struct}$.*

**Expression 5.6** (Epistemic part). *The training-induced component satisfies a vanishing bound*

$$\text{CoD}_D(T) \leq \frac{C_{\text{ep}}}{1-\gamma}\sqrt{\frac{d_{\text{eff}}\log(T/\delta)}{T}} + \frac{b^\star}{1-\gamma},$$

*with $b^\star = 0$ under exact realizability, so $\text{CoD}_D(T) \to 0$ as $T \to \infty$.*

**Noise floor.** Exogenous process and observation noise contribute an additive term $C_{\text{noise}}/(1-\gamma)$ that is persistent and purely environmental.

## 6 EXPERIMENT

We design a one-shot content-moderation delegation task that mirrors the helpfulness–safety toy model in Section 4.3, using RealToxicityPrompts (Gehman et al., 2020) with a Qwen3 policy model and Qwen3Guard safety model (Yang et al., 2025; Zhao et al., 2025). For each prompt $x_i$ and action $a \in \{\text{ACCEPT}, \text{REWRITE}, \text{BLOCK}\}$, Qwen3 produces a candidate response scored by

$$r_i(a; \lambda) = H_i(a) - \lambda S_i(a),$$

where $H_i(a) \in [0, 1]$ is a helpfulness score and $S_i(a) \in [0, 1]$ is a risk score from Qwen3Guard; the safety weight $\lambda$ is our ablation knob. Let $J_{\text{oracle}}(\lambda)$ be the average reward of per-sample maximizers of $r_i(a; \lambda)$, $J_\ell^*(\lambda)$ the best achievable with level-$\ell$ signals, and $\text{CoD}_\ell(\lambda) = J_{\text{oracle}}(\lambda) - J_\ell^*(\lambda)$ the corresponding empirical information cost of delegation.

We bucket prompts into benign, borderline, and toxic groups using toxicity scores. Qwen3Guard outputs a ternary safety label (Safe / Controversial / Unsafe) and a fine-grained risk category. We define three information levels: $L1$ uses a binary signal $g_i^{(1)}(a) \in \{\text{SAFE}, \text{UNSAFE}\}$ by merging CONTROVERSIAL into UNSAFE; $L2$ uses the full ternary label $g_i^{(2)}(a) \in \{\text{SAFE}, \text{CONTROVERSIAL}, \text{UNSAFE}\}$; $L3$ uses the pair $g_i^{(3)}(a) = (\text{label}, \text{top category})$. Since $L1$ and $L2$ are deterministic coarsenings of $L3$, they satisfy $L3 \succeq L2 \succeq L1$ in Blackwell's order.

Figure 2 (left) reports mean rewards by toxicity bucket and action. For benign prompts, ACCEPT and REWRITE dominate BLOCK; for toxic prompts, BLOCK is optimal and ACCEPT performs poorly, with borderline prompts in between. Thus the reward landscape is clearly structured in the (bucket, action) space, so safety-type information can genuinely change optimal actions, consistent with the decision-sensitive notion of information value in Section 5.2. Figure 2 (right) plots $\text{CoD}_\ell(\lambda)$ for $\ell \in \{L1, L2, L3\}$. We observe $\text{CoD}_\ell(\lambda) > 0$ for all $\lambda \geq 0$, including $\lambda = 0$, since acting only through compressed guard signals cannot match oracle performance even when only $H$ matters. Moreover, $\text{CoD}_{L1}(\lambda)$ and $\text{CoD}_{L2}(\lambda)$ are nearly identical, while $\text{CoD}_{L3}(\lambda)$ is uniformly

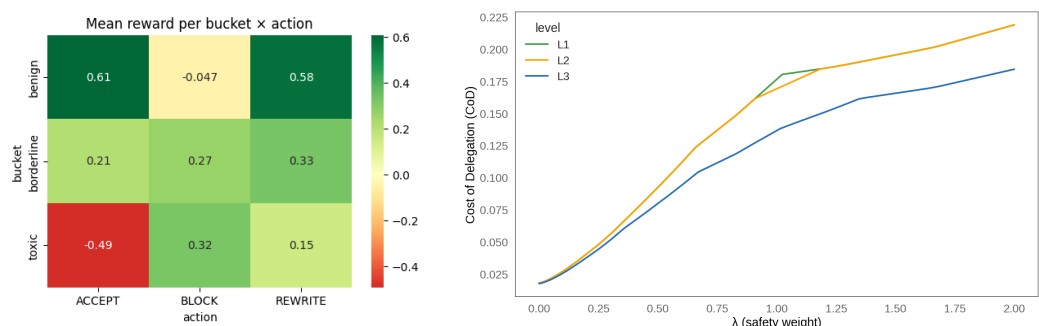

Figure 2: **Left**: reward landscape. **Right**: CoD vs safety weight.

smaller: splitting CONTROVERSIAL from UNSAFE (L1→L2) adds label entropy but little decision value, whereas adding categories (L2→L3) refines signals exactly where optimal actions differ, reducing the decision-relevant information cost $\Delta_{info}$. All three curves increase with $\lambda$, and the gap between $L1/L2$ and $L3$ widens as safety becomes more heavily weighted, matching our structural analysis that sharper safety curvature amplifies information-structural costs.

## 7    RELATED WORK AND IMPLICATIONS FOR MODERN SYSTEMS

Reward modeling and RLHF align language models with human preferences (Ouyang et al., 2022; Rafailov et al., 2023). Recent work probes robustness, distribution shift and benchmarked evaluation of reward models (Lambert et al., 2025; Shao et al., 2024) and analyzes length bias in preference-based training (Park et al., 2024). Our results suggest that, beyond lowering generic reward error, it is crucial to allocate modeling capacity to those preference distinctions that actually change optimal policies. In CoD terms, this is an information-structure question: reward models that collapse decision-irrelevant variance but sharpen decision-relevant boundaries can substantially reduce $\Delta_{info}$ even when global fit is imperfect.

Mixture-of-experts architectures scale capacity by sparsely activating experts (Shazeer et al., 2017; Fedus et al., 2021; Mustafa et al., 2022; Huang et al., 2024a; Qiu et al., 2025). Existing routing rules are mostly variance- or similarity-based. Under our framework, sparsity interacts with CoD through $\Delta_{coord}$: experts are most useful when they partition state–task space along decision-sensitive directions, not high-variance but policy-irrelevant axes. This view is consistent with evidence that task-aware routing and specialization matter more than raw parameter count.

Reasoning models that expand test-time computation via chain-of-thought or RL-trained scratch-pads (Wei et al., 2022; Wang & Zhou, 2024; Guo et al., 2025; OpenAI, 2024) can likewise be read through CoD: they dynamically enrich the principal's observation and policy classes, mainly reducing information and surrogate gaps rather than eliminating delegation itself. Further justification and additional empirical connections to modern systems are provided in Appendix A.

## 8    CONCLUSION

Our work provides a new perspective and unified framework for understanding why perfect alignment remains elusive even with abundant data and computational resources. We show that information value is decision-theoretic, only directions that change optimal actions matter. This suggests that scalable alignment must combine target design (what is optimized) with structural design (who observes what, and how they interact), and that complex systems should be judged not only by loss, but by the size and shape of the structural gaps they induce. Our analysis is limited by surrogates and simplified tasks, but it motivates future work that turns CoD into concrete design insights.

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

APPENDIX

## A  MORE RELATED WORK

Our analysis sits at the intersection of classical team decision theory, decentralized control, dynamic game theory, and contemporary AI alignment. At a high level, we adopt the team-theoretic lens that views a collection of agents with (nominally) common objectives but heterogeneous information, and we ask not for exact optimal policies (which are often intractable), but for quantitative bounds on the *structural* cost of delegation. The main novelties are: (i) a four-level hierarchy that disentangles delegation, coordination sparsity, and information structure; and (ii) an explicit decomposition of the Cost of Delegation into $\Delta_{\text{deleg}}$, $\Delta_{\text{coord}}$, and $\Delta_{\text{info}}$ within an LQ surrogate, which can be related to Blackwell's ordering of information structures.

**Team decision theory and decentralized LQ control.**  Classical team decision theory analyzes stochastic control problems with multiple decision makers sharing a common payoff but observing different signals (Radner, 1962; Marschak & Radner, 1958). This line of work established foundational concepts such as team-optimality and the role of information patterns, but typically did not provide explicit formulas for the performance gap between centralized and decentralized solutions. Ho and Chu systematically classified information structures and identified conditions (such as partial nestedness) under which person-by-person optimality implies team optimality (Ho & Chu, 1972). In decentralized LQG, Sandell and Athans showed that nonclassical information patterns can make the optimal controller highly nontrivial even in linear-quadratic settings (Sandell & Athans, 2003), echoing the Witsenhausen counterexample. Athans later surveyed decentralized control architectures and emphasized that structural constraints, rather than noise, often dominate performance limits (Athans, 1975).

Our framework is close in spirit but different in emphasis. We do not attempt to compute optimal decentralized policies for a given information pattern. Instead, we introduce a surrogate LQ world in which the centralized optimum is explicitly computable, and we then quantify how much performance is lost as one moves from a centralized controller (L1) to a multi-principal Nash equilibrium (L2), then to sparse coordination (L3), and finally to partial observation (L4). In that sense, our results are complementary to the structural existence results of Radner (1962); Ho & Chu (1972): we treat information and coordination patterns as design objects and attach explicit performance penalties to them.

**Information structures and Blackwell ordering.**  Blackwell's comparison of experiments formalizes when one information structure is more informative than another in a decision-theoretic sense (Blackwell, 1953). Recent work on information structure design revisits this question for team problems and team games, asking how to add or rewire information links to improve performance (Summers et al., 2017). Our notion of an "information cost" is directly in this tradition. In the LQ surrogate, $\Delta_{\text{info}}$ can be written in terms of conditional covariance operators and is monotone with respect to Blackwell dominance of the projections. However, unlike most of the classical literature, we explicitly separate information structure from coordination structure, even under a fixed information pattern, sparsifying the coordination matrix $W$ induces a distinct cost component $\Delta_{\text{coord}}$.

Nayyar et al. (2013) show that certain decentralized stochastic control problems with partial history sharing can be reformulated as centralized POMDPs using a "common information" state, which restores dynamic programming. Their goal is to recover tractable dynamic programming recursions. Our goal, in contrast, is quantitative. we keep a fixed LQ surrogate and use it to decompose the performance gap between centralized and decentralized architectures into mechanism-design, coordination, and information components. The two perspectives are compatible: the common-information state can be viewed as an extreme point in the lattice of information structures, and our bounds describe how far a given architecture lies from that ideal.

**Dynamic games and sparse coordination.**  Our simultaneous-move LQ game between principals is related to the literature on dynamic and differential games (Başar & Olsder, 1999), in which agents optimize individual quadratic costs subject to linear dynamics, and Nash equilibria can often be characterized in closed form. However, most of that literature either assumes relatively dense

coupling or focuses on stability and solvability rather than explicit performance decompositions. In parallel, coordination graphs and factored multi-agent models aim to exploit sparse interaction structure to scale planning and RL (Guestrin et al., 2002). Our coordination matrix $W$ can be seen as a continuous analogue of such graphs, but our focus is inverted. Rather than using sparsity to *design* scalable algorithms, we ask how much performance is lost when sparsity is imposed as a constraint.

**Principal–agent alignment and cooperative assistance games.** Within AI alignment, Hadfield-Menell's line of work argues that principal–agent misalignment provides a more realistic model for AI systems than idealized single-agent optimization (Hadfield-Menell et al., 2016; Hadfield-Menell, 2021). CIRL formalizes value alignment as a cooperative partial-information game in which the human knows the reward and the robot must infer it; the principal-agent thesis systematizes this view and emphasizes strategic behavior by both sides. Our contribution is orthogonal. We *assume* the principal's objective is given (up to surrogate approximation) and ask how much loss is unavoidable purely because control is delegated to multiple principals with limited coordination and information. In this sense, our Cost of Delegation framework provides quantitative tools inside the principal–agent paradigm, it gives explicit upper and lower bounds on the gap between centralized optimal control and the behavior of a delegated, structured system.

**Reward models, preference optimization, and modern alignment.** A large body of recent work studies alignment via preference-based reward modeling and policy optimization, including RL from human preferences (Christiano et al., 2017), RLHF for summarization and instruction-following (Stiennon et al., 2020; Ouyang et al., 2022), and direct preference optimization methods that re-interpret policies as implicit reward models (Rafailov et al., 2023). Constitutional AI further emphasizes structured feedback and safety constraints (Bai et al., 2022). These methods primarily target the *reward-specification* problem, learning a reward or preference model that reflects human judgment. Our decomposition is complementary, even if the reward were perfectly specified, delegation to multiple principals, sparse coordination, and partial observation induce a residual Cost of Delegation.

In our framework, the relevant notion of information value is decision-theoretic in nature: what matters is how observations change optimal actions, not how much entropy they carry. High-variance but decision-independent features can be essentially useless, while low-variance but decision-sensitive directions dominate delegation cost. This perspective reframes several alignment problems under a single CoD lens. It suggests that reward models should prioritize distinctions that actually move the policy, and that architectural choices (such as which internal signals to expose to which modules) should be evaluated by their impact on optimal actions rather than their raw information content. Further implications for modern systems (including mixture-of-experts routing and multi-agent orchestration) are discussed in the main text and in subsequent appendices.

## B  IMPLICATIONS FOR MODERN SYSTEMS

Our framework reframes several alignment problems under a single Cost-of-Delegation (CoD) lens.

**Reward models and RLHF-style training.** RLHF and related preference-learning schemes train reward models to approximate human judgments over model outputs, which are then used to optimize policies via RL or direct preference optimization (Christiano et al., 2017; Stiennon et al., 2020; Ouyang et al., 2022; Bai et al., 2022; Rafailov et al., 2023). Recent evaluations show that current reward models are often miscalibrated and brittle across tasks (Lambert et al., 2025). A particularly robust finding is *length bias*, that rewards correlate strongly with response length even when humans do not, which can drive systematic reward hacking and degenerate behaviors (Singhal et al., 2023; Huang et al., 2024b).

Our information-cost expression writes the gap between full and lossy observations as

$$\Delta_{\text{info}} = \frac{1}{2}\,\mathbb{E}[\delta u^\top S_k \delta u] = \frac{1}{2}\,\text{tr}\big(G_k^{-T} S_k G_k^{-1}\,\text{Cov}(\epsilon)\big) = \frac{1}{2}\,\text{tr}\big(G_k^{-T} S_k G_k^{-1}\, L\Sigma_\phi L^\top\big).$$

where $\tilde{K}_m$ encodes how state directions affect value gradients and $\Sigma_{mj}^\perp$ is the residual covariance that remains invisible under the chosen observation structure. In this view, an observation or feature is valuable only insofar as it reduces *decision-relevant* residual variance along directions weighted by $\tilde{K}_m$, in line with Blackwell's comparison of experiments (Blackwell, 1953). High-entropy features that barely move the optimal action leave $\Sigma_{mj}^\perp$ essentially unchanged, and thus do not reduce $\Delta_{\text{info}}$, even if they explain a large fraction of raw outcome variance.

Empirical findings on reward models can be interpreted through this lens. Length and other stylistic proxies typically explain large variance in human scores but have low marginal effect on the ranking of candidate actions in safety-critical regions (Singhal et al., 2023; Lambert et al., 2025; Huang et al., 2024b). In CoD terms, they primarily reshape reward level sets away from the decision boundary and therefore contribute little to closing the centralized–decentralized gap. Our toy content-moderation experiment mirrors this by showing that moving from a coarse binary risk label (L1) to a slightly higher-entropy threeway label (L2) barely changes CoD, while adding structured category information (L3) that separates prompts requiring different actions substantially reduces $\Delta_{\text{info}}$. This suggests that reward models should devote capacity to partitioning the space along those preference distinctions that actually switch optimal actions, rather than uniformly modeling all preference variation. Recent proposals that disentangle quality and stylistic or length signals, or that perform post-hoc calibration of reward models, can be read as attempts to reduce decision-irrelevant components of $\Sigma^\perp$ while preserving decision-relevant gradients (Singhal et al., 2023; Huang et al., 2024b).

**Mixture-of-experts and sparse coordination.** Mixture-of-experts (MoE) architectures implement sparse routing of tokens to experts, effectively choosing a sparse coordination matrix between submodules (Shazeer et al., 2017; Fedus et al., 2021; Mustafa et al., 2022). Large MoE models for vision and language have shown strong scaling properties, but also exhibit issues such as expert collapse, unbalanced routing, and specialization on redundant features (Cai et al., 2025; Gan et al., 2025). Much of the design effort focuses on router objectives and regularizers that encourage load balancing and diversity of experts (Zhou et al., 2022; Gupta et al., 2022).

In our notation, replacing an ideal dense coordination matrix $W^*$ with a sparse $W_k$ induces a coordination gap

$$\Delta_{\text{coord}} \lesssim \frac{1}{m^2} \|G_k^{-1}\|_2^2 \|W^* - W_k\|_F^2 \|K\|_2^2,$$

so the performance loss depends not just on how sparse $W_k$ is, but on *which* couplings are dropped relative to the true interaction structure $W^*$. From this angle, MoE routing is a particular mechanism for choosing $W_k$ as conventional token-level routers tend to cluster tokens by similarity in representation space, which correlates more with statistical variance than with the task-specific influence of those tokens on downstream losses (Cai et al., 2025). Our bound suggests that sparsity is benign when it respects the underlying "coordination graph" encoded by $W^*$, but costly when it severs edges along directions with large $K$ or strong cross-expert couplings.

Recent work on multi-task and task-aware MoEs implicitly moves in this direction, designing routing objectives that align expert assignment with gradient structure or task identity rather than pure feature clustering (Gupta et al., 2022; Cai et al., 2025). Under the CoD lens, such methods can be interpreted as choosing sparsity patterns that minimize $\|W^* - W_k\|_F$ along decision-critical directions, thereby reducing $\Delta_{\text{coord}}$ while retaining most of the computational benefits of sparse activation. Low-rank adaptation methods such as LoRA (Hu et al., 2022) can similarly be viewed as constrained perturbations of $W^*$ and $R$, whose effect on CoD is governed by how the low-rank updates interact with the value gradients encoded in $K$ and $\tilde{K}_m$.

**Reasoning models and process-level supervision.** Chain-of-thought prompting and process supervision augment models with intermediate reasoning trajectories that are explicitly scored or constrained (Wei et al., 2022; Wang & Zhou, 2024). Recent reasoning-centric models such as DeepSeek-R1 and OpenAI's o1 scale this idea further by combining RL with carefully designed reward signals and process data to incentivize extended reasoning and self-checking behavior (Guo

et al., 2025; OpenAI, 2024). Conceptually, these approaches expand the effective state and observation spaces. The system observes not only the external query but also a rich internal trace of tentative computations, tool calls, and justification.

Within our framework, such traces can be viewed as additional observation channels that reduce $\Sigma^\perp$ along directions that strongly affect final decisions. Process-level supervision shapes these channels so that the internal trajectories are informative about correctness and safety, not just about superficial fluency. In other words, the extra bits are useful because they align with value gradients as they make it easier for a principal (the outer RL loop, a verifier, or a downstream orchestrator) to distinguish between candidate actions that would otherwise look similar at the surface level. This perspective complements existing accounts of reasoning models as implementing longer computation or better search by emphasizing the informational role of scratchpads and verification signals in shrinking $\Delta_{\text{info}}$ rather than merely increasing model capacity.

**Multi-agent orchestration and tool ecosystems.** Finally, multi-agent LLM systems and tool ecosystems instantiate delegation at the system level. Planners, solvers, critics, retrievers, and external tools act as distinct delegates whose interactions are mediated by prompts, APIs, and routing policies. Our four-level hierarchy provides a coarse template for such designs. Co-locating capabilities in a single monolithic model corresponds to L1; introducing specialized agents corresponds to L2; enforcing sparse communication or rigid workflows corresponds to L3; restricting agents to partial context windows or filtered observations corresponds to L4. The CoD decomposition then clarifies which inefficiencies are fundamentally architectural (e.g., unavoidable losses from partial observability or strict modularization) and which are amenable to better objective design or training.

Taken together, these connections suggest that CoD does not propose yet another specific alignment algorithm, but rather offers a unifying language for analyzing why different modern systems succeed or fail. Across reward modeling, MoE routing, reasoning models, and multi-agent orchestration, the same lesson repeats. What matters is not how many bits we observe or how many parameters we add, but how strongly those design choices couple to the directions in state space that actually move optimal actions.

## C  FULL STATEMENT OF THE FRAMEWORK

In this section we state the full framework, which covers contents in Section 3 and 4.1. Some expressions are stated again for logic consistency. For proofs, see the next section.

### C.1  PRELIMINARIES

**Notation and probability space.** Fix integers $k \geq 1$ and block sizes $M_1, \ldots, M_k \geq 1$ with $M := \sum_{i=1}^k M_i$ total delegates, where delegate $m \in \{1, \ldots, M\}$ belongs to block $i(m) \in \{1, \ldots, k\}$. Let $(\Omega, \mathcal{F}, \mathbb{P})$ be a probability space with filtration $(\mathcal{F}_t)_{t \geq 0}$.

**State and action spaces.** The system state $\phi_t \in \mathcal{S} \subseteq \mathbb{R}^n$ evolves on measurable set $\mathcal{S}$. The principal chooses $u_{P,t} \in \mathcal{U}_P$ (compact subset of $\mathbb{R}^{d_P}$). Each delegate $m$ selects $u_{m,t} \in \mathbb{R}^d$, stacked as $\mathbf{u}_t = (u_{1,t}^\top, \ldots, u_{M,t}^\top)^\top \in \mathbb{R}^{Md}$. The aggregate action is

$$a_t = \sum_{m=1}^M u_{m,t} \in \mathbb{R}^d, \tag{1}$$

with operator norm $\|\mathcal{A}\|_2 = \sqrt{M}$,[2] which appears explicitly in stability and learning bounds.

---

[2] The aggregation operator $\mathcal{A} : \mathbb{R}^{Md} \to \mathbb{R}^d$ has matrix representation $[I_d \ I_d \ \cdots \ I_d]$. Throughout, $\|\cdot\|_2$ denotes Euclidean/spectral norm; $\rho(\cdot)$ denotes spectral radius.

## C.2 DYNAMICS, INFORMATION, AND LEARNING

**Dynamics.** The controlled dynamics are

$$\phi_{t+1} = f(\phi_t, a_t) + w_t, \tag{2}$$

where $f : \mathcal{S} \times \mathbb{R}^d \to \mathcal{S}$ is the deterministic transition map and $\{w_t\}_{t \geq 0}$ is process noise.

---

**Assumption S1** (MDP Regularity and Lyapunov Drift)**:**

(a) *Lipschitz dynamics.* There exists $L_f \geq 0$ such that for all $(s, a), (s', a') \in \mathcal{S} \times \mathbb{R}^d$,

$$\|f(s, a) - f(s', a')\|_2 \leq L_f (\|s - s'\|_2 + \|a - a'\|_2).$$

If $f$ is twice differentiable, then $\sup_{(s,a) \in \mathcal{S} \times \mathbb{R}^d} \|\nabla^2 f(s, a)\|_2 \leq H_f < \infty$.

(b) *Sub-Gaussian noise.* $\{w_t\}_{t \geq 0}$ forms a martingale difference sequence: $\mathbb{E}[w_{t+1}|\mathcal{F}_t] = 0$ a.s. Moreover, there exists $\sigma_w > 0$ such that for all deterministic $u \in \mathbb{R}^n$ and all $t \geq 0$,

$$\mathbb{E}\left[\exp(u^\top w_{t+1})|\mathcal{F}_t\right] \leq \exp\left(\frac{\sigma_w^2}{2}\|u\|_2^2\right) \quad \text{a.s.}$$

(c) *Bounded domain and actions.* The state space $\mathcal{S}$ is compact with $\sup_{\phi \in \mathcal{S}} \|\phi\|_2 \leq B_\phi$. Each delegate's action satisfies $\|u_m\|_2 \leq B_u$, giving aggregate bound $\|a_t\|_2 \leq MB_u$. Define the operating set $\mathcal{S}_{\text{op}} := \mathcal{S}$.

---

**Information.** Asymmetric partial observability: principal receives noisy aggregate feedback while delegates observe only local state components and neighbor actions.

---

**Assumption S2** (Observation)**:** Principal observes signals through three channels:

(a) *State:* $\tilde{\phi}_t = \phi_t + \xi_t \in \mathbb{R}^n$.
(b) *Response:* $y_t = \Psi(\mathbf{u}_t) + \zeta_t \in \mathbb{R}^{d_y}$, where $\Psi : \mathbb{R}^{Md} \to \mathbb{R}^{d_y}$ is $L_\Psi$-Lipschitz.
(c) *Reward:* $r_t = r(\phi_t, a_t, u_{P,t}) + \varepsilon_t \in \mathbb{R}$, where $r : \mathcal{S} \times \mathbb{R}^d \times \mathcal{U}_P \to [-R_{\max}, R_{\max}]$.

---

The noise processes $\xi_t, \zeta_t, \varepsilon_t$ are $\mathcal{F}_t$-adapted martingale difference sequences with $\mathbb{E}[\cdot|\mathcal{F}_t] = 0$. Each is conditionally sub-Gaussian with parameters $\sigma_\xi^2, \sigma_\zeta^2, \sigma_r^2$ respectively. The concatenated noise $\eta_t := (\xi_t^\top, \zeta_t^\top, \varepsilon_t)^\top \in \mathbb{R}^{n+d_y+1}$ satisfies $\mathbb{E}[\exp(v^\top \eta_{t+1})|\mathcal{F}_t] \leq \exp(\frac{1}{2} v^\top \Sigma_\eta v)$ for some $\Sigma_\eta \succeq 0$ and all deterministic $v$, allowing cross-channel correlations.

---

**Assumption S3** (Information Architecture)**:**

(a) *Principal's information:* At time $t$, observes history $\mathcal{H}_{P,t} = \{\tilde{\phi}_s, y_s, r_s, u_{P,s}\}_{s=0}^{t-1} \cup \{\tilde{\phi}_t\}$ and chooses $u_{P,t}$ measurably with respect to $\mathcal{H}_{P,t} \subseteq \mathcal{F}_t$.

(b) *Delegate's local observation:* Delegate $m$ observes $\phi_{m,t} = \Pi_m \phi_t + \nu_{m,t}$ where $\Pi_m \in \mathbb{R}^{n_m \times n}$ with $\|\Pi_m\|_2 \leq 1$. The collective observation satisfies $\sum_{m=1}^M \Pi_m^T \Pi_m \preceq \kappa I_n$ for some $\kappa < \infty$. The noise $\nu_{m,t}$ follows S2's sub-Gaussian structure with parameter $\sigma_\nu^2$. Each delegate observes $u_{P,t}$.

---

**Learning.** Principal cannot observe individual actions or reconstruct the aggregate action $a_t$ from the response signal $y_t = \Psi(\mathbf{u}_t) + \zeta_t$. Instead, he learns reduced-form predictive models:

$$\mathcal{M}_\phi : \mathbb{E}[\tilde{\phi}_{t+1}|\tilde{\phi}_t, u_{P,t}] = F_\theta(\tilde{\phi}_t, u_{P,t}), \quad \mathcal{M}_r : \mathbb{E}[r_{t+1}|\tilde{\phi}_t, u_{P,t}] = R_\theta(\tilde{\phi}_t, u_{P,t}), \tag{3}$$

where $\theta = (\theta_\phi, \theta_r)$ parameterizes the reduced-form predictors. For realizability we assume $F_\theta$ and $R_\theta$ belong to linear-in-parameters classes. Define the predictable regressor:

$$\bar{X}_t := \begin{bmatrix} \tilde{\phi}_{t-1}^\top & u_{P,t-1}^\top & 1 \end{bmatrix}^\top \in \mathbb{R}^{n+d_P+1} \tag{4}$$

which is $\mathcal{H}_{P,t-1}$-measurable. Principal uses the observable pairs $\{(\bar{X}_t, \tilde{\phi}_t, r_t)\}_{t=1}^T$ for estimation.

**Assumption S4** (Persistent Excitation)**:** On the operating set $\mathcal{S}_{\mathrm{op}}$ from S1(c):

(a) *Boundedness:* For any horizon $T$ and confidence $\delta \in (0,1)$,

$$\Pr\left[\max_{1 \leq t \leq T} \|\bar{X}_t\|_2 \leq B_X(T,\delta)\right] \geq 1 - \delta$$

where $B_X(T,\delta) = O(\sqrt{\log(T/\delta)})$ under S2.

(b) *Sliding-window excitation:* The principal's policy ensures that for all $t \geq T_0$:

$$\lambda_{\min}\left(\frac{1}{T_0}\sum_{s=t-T_0+1}^{t}\bar{X}_s\bar{X}_s^\top\right) \geq \alpha$$

**Principal objective.** Hierarchical uncertainty from system noise $(w_t)$, partial observations $(\xi_t, \zeta_t, \varepsilon_t)$, and indirect control through delegate equilibrium motivate risk-sensitive objectives:

$$J^{\pi_P}(\phi_0) = U_\beta\left[\sum_{t=0}^{T-1}\gamma^t r(\phi_t, a_t, u_{P,t})\right], \tag{5}$$

where $\gamma \in (0,1)$ is the discount factor, $U_\beta(X) := -\mathrm{CVaR}_\beta(-X)$ is the coherent risk-sensitive utility with $\beta \in (0,1)$ controlling risk aversion, and aggregate action $a_t = \sum_{m=1}^{M} u_{m,t}^*$ results from the delegate Nash equilibrium. $U_\beta$ admits a tractable *minimax* formulation (the derivation is standard) under the subsequently introduced G1-G2, ensuring that hierarchical learning remains computationally feasible despite risk considerations.[3]

**Certainty Equivalence Approximation.** Delegates act on state estimates as if certain, reducing the POMDP to an LQG surrogate. This approximation is exact under Gaussian noise and provides controlled error under S2's sub-Gaussian structure. The following analysis derives the induced game structure under this approximation.

C.3    DELEGATE GAME FROM FIRST PRINCIPLES

**Delegate objective.** Given S3's information structure, delegate $m$ minimizes:

$$\mathbb{E}\left[\sum_{t=0}^{\infty}\gamma^t\left(\frac{1}{2}\|\Pi_m\phi_t - \psi_m(u_{P,t})\|_{Q_m}^2 + \frac{1}{2}\|u_m\|_{R_m}^2\right)\right] \tag{6}$$

where $\psi_m(u_P) = \bar{\phi}_m^* + P_m u_P$ is the principal-influenced target.

**Quadratic Approximation.** We adopt certainty equivalence and linearize $f(\phi, a)$ at $(\bar{\phi}, 0)$ while quadraticizing the tracking losses at $z = 0$, obtaining a quadratic surrogate stage game.

**Lemma C.1** (Local linearization). *Fix $\bar{\phi} \in \mathcal{S}_{\mathrm{op}}$. There exist $A := D_\phi f(\bar{\phi}, 0)$, $B := D_a f(\bar{\phi}, 0)$ and $c > 0$ such that $f(\phi, a) = f(\bar{\phi}, 0) + A(\phi - \bar{\phi}) + Ba + R_2(\phi, a)$ with $\|R_2(\phi, a)\| \leq c(\|\phi - \bar{\phi}\|^2 + \|a\|^2)$ for all $(\phi, a) \in \mathcal{S}_{\mathrm{op}} \times \{\|a\| \leq MB_u\}$.*

**Theorem C.1** (Emergent coordination). *Under certainty equivalence and local linearization, the infinite-horizon problems reduce (to second order) to the single-stage game*

$$J_m(u_m; u_{-m}) = \frac{1}{2}u_m^T R_m u_m + u_m^T K_m(\delta\phi, u_P) + \frac{1}{2}\sum_{j=1}^{M}u_m^T W_{mj}u_j + \varepsilon_m(\delta\phi, \mathbf{u}), \tag{7}$$

*with $\delta\phi := \phi - \bar{\phi}$, $K_m = (\Pi_m B)^T Q_m[(\Pi_m A)\delta\phi - P_m u_P]$ and $W_{mj} = (\Pi_m B)^T Q_m(\Pi_m B)$.*

*The approximation error $\varepsilon_m$ satisfies:*

$$|\varepsilon_m(\delta\phi, \mathbf{u})| \leq C_h^{(m)}\|\delta\phi\|_2^3 + C_f^{(m)}(\|\delta\phi\|_2^2 + \|\mathbf{a}\|_2^2), \tag{8}$$

---

[3]All expectations are under $\mathbb{P}^{\pi_P}$, the law induced by policy $\pi_P$, S1-S3, and the equilibrium mapping.

*where $C_h^{(m)}$ bounds the cubic remainder from value function approximation and $C_f^{(m)}$ bounds the quadratic remainder from dynamics linearization.*

See Appendix A for full derivation. The coordination matrix $W$ emerges mechanistically: delegate $j$'s action propagates through the shared dynamics $B$ to affect delegate $m$'s future states via projection $\Pi_m$. To formalize, $W = [W_{mj}]$ induces the physical graph $\mathcal{G} = (V, E)$ with $V = \{1, \ldots, M\}$ and $E = \{(m, j) : W_{mj} \neq 0\}$, which is in general distinct from the observation structure in S3.

**Equilibrium.** Each delegate $m$'s surrogate objective (Theorem C.1) depends on other delegates' actions through the coupling terms $\sum_j u_m^T W_{mj} u_j$. This creates strategic interdependence: delegate $m$'s optimal choice $u_m^*$ depends on the profile $u_{-m}$ of all other delegates. To characterize the equilibrium, we stack all delegate decisions $\mathbf{u} = (u_1^T, \ldots, u_M^T)^T$ into a single strategic game.

**Proposition C.1** (Nash Equilibrium). *The aggregate game*

$$J(\mathbf{u}; \delta\phi, u_P) = \frac{1}{2}\mathbf{u}^T(R + W)\mathbf{u} + \mathbf{u}^T K(\delta\phi, u_P) \tag{9}$$

*has unique Nash equilibrium $\mathbf{u}^* = -G^{-1}K(\delta\phi, u_P)$ when $G := R + W$ satisfies G1(a) below. For the full system, the equilibrium map is $\mathbf{u}^*(\phi, u_P) := -G^{-1}K(\phi - \bar{\phi}, u_P)$, which is Lipschitz with constant $\|G^{-1}\|_2 L_K$ in $(\delta\phi, u_P)$.*

Direct computation requires $O((Md)^3)$ operations, so we impose:

---

**Assumption G1** (Structured Coordination)**:**

(a) *Well-posedness:* $\lambda_{\min}((G + G^T)/2) \geq m > 0$
(b) *Bounded coupling:* $\|W\|_2 \leq w_{\max} < \infty$
(c) *Tractable structure:* $W$ admits one of:
   - *Low-rank + sparse:* $W = UV^T + S$ with $U, V \in \mathbb{R}^{Md \times r}$, $r \ll Md$, $S$ sparse
   - *Tree/DAG:* Sparsity follows directed acyclic or tree structure
   - *Block-sparse:* At most $k$ nonzero blocks per row

---

**Proposition C.2** (Equilibrium Properties). *Under G1, the Nash equilibrium $\mathbf{u}^*$:*

*(a)* Is Lipschitz continuous with constant $L_K/m$ in $(\phi, u_P)$

*(b)* Solves the first-order condition $(R + W)\mathbf{u} = -K(\phi - \bar{\phi}, u_P)$

*(c)* For symmetric $W$, minimizes the potential $\Phi(\mathbf{u}) = \frac{1}{2}\mathbf{u}^T G\mathbf{u} + \mathbf{u}^T K(\phi - \bar{\phi}, u_P)$

*Remark.* On compact $\mathcal{S} \times \mathcal{U}_P$ with continuous $K$, choosing $B_u \geq \sup_{(\phi, u_P)} \|G^{-1}K(\phi - \bar{\phi}, u_P)\|_\infty$ ensures the unconstrained equilibrium respects action bounds.

**Closed-loop Error Propagation.** The equilibrium $\mathbf{u}^*(\phi, u_P) = -G^{-1}K(\phi - \bar{\phi}, u_P)$ computed from the quadratic surrogate is substituted into the *true* dynamics:

$$F(\phi; u_P) = f(\phi, \mathcal{A}(\mathbf{u}^*(\phi, u_P))). \tag{10}$$

Using the surrogate-based equilibrium $\mathbf{u}^*$ instead of the infinite-horizon optimal actions requires closed-loop contractivity for bounded cumulative propagation. Under closed-loop contraction with $L_{cl} := \sup \|D_\phi F(\phi; u_P)\|_2 < 1/\gamma$, the discounted cumulative error from surrogate-based equilibrium satisfies:

$$\sum_{t=0}^{\infty} \gamma^t |\text{error}_t| = O\left(\frac{C_h + C_f}{1 - \gamma L_{cl}}\right) \tag{11}$$

where $C_h$ controls the cubic (value function) error and $C_f$ controls the quadratic (dynamics) error.

> **Assumption G2** (Stability)**:** The closed-loop Jacobian satisfies
> $$\sup_{(\phi, u_P) \in \mathcal{S} \times \mathcal{U}_P} \|D_\phi F(\phi; u_P)\|_2 < \frac{1}{\gamma} \tag{12}$$
> where $F(\phi; u_P) = f(\phi, \mathcal{A}(\mathbf{u}^*(\phi, u_P)))$. Under compact domains, it is finite and verifiable.

**Theorem C.2** (Closed-loop contraction)**.** *Under G1 and G2's sufficient condition*

$$L_s + L_a \|\mathcal{A}\|_2 \|G^{-1}\|_2 L_K < \frac{1}{\gamma}, \tag{13}$$

*the closed-loop system satisfies* $\sup \|D_\phi F\|_2 < 1/\gamma$. *Moreover, the cumulative approximation error from using the quadratic surrogate remains bounded.*

# D    APPENDIX FOR SECTION 3 AND 4.1 (APPENDIX C)

## D.1    REMARKS FOR S1

**(a) Lipschitz & smoothness.**   Assume $f : \mathcal{S} \times \mathbb{R}^d \to \mathcal{S}$ is globally $L_f$–Lipschitz in $(s, a)$ and $C^2$ on $\mathcal{S} \times \{a : \|a\| \le MB_u\}$ (the action bound is from S1(c)). Then the Jacobians

$$A(s, a) := D_s f(s, a)$$

and

$$B(s, a) := D_a f(s, a)$$

exist a.e., with

$$\|A(s, a)\| \le L_s, \quad \|B(s, a)\| \le L_a$$

for finite $L_s, L_a$ used in G2.

**(b) Sub-Gaussian MDS.**   For $\{w_t\}$, the conditional MGF bound

$$\mathbb{E}[\exp(u^\top w_{t+1}) \mid \mathcal{F}_t] \le \exp(\tfrac{1}{2}\sigma_w^2 \|u\|^2)$$

implies:

(i) $\mathbb{E}[w_{t+1} \mid \mathcal{F}_t] = 0$ a.s.;

(ii) vector-valued Freedman/Azuma inequalities hold uniformly over directions $u$;

(iii) stability and concentration results are *dimension free* up to log factors.

**(c) Compact operating set.**   We take $\mathcal{S}$ compact with $\sup_{\phi \in \mathcal{S}} \|\phi\| \le B_\phi$ and per-delegate action bound $\|u_m\| \le B_u$, hence $\|a_t\| \le MB_u$ and all linearizations are invoked on a compact set. This replaces a Lyapunov drift assumption and suffices for the local Taylor bounds used below.

## D.2    REMARKS FOR S2

**(a) Channels and joint noise.**   Each channel noise is an $\mathcal{F}_t$–adapted MDS with sub-Gaussian proxy $\sigma_\eta$. When cross-channel correlations are present, the concatenated $\eta_t := (\xi_t^\top, \zeta_t^\top, \varepsilon_t)^\top$ satisfies

$$\mathbb{E}[\exp(v^\top \eta_{t+1}) \mid \mathcal{F}_t] \le \exp(\tfrac{1}{2} v^\top \Sigma_\eta v)$$

with some PSD $\Sigma_\eta$, enabling joint self-normalized bounds.

**(b) Measurability.**   All observation maps $(\Psi, r)$ are Borel and the histories $\mathcal{H}_{P,t}$ are $\sigma$–fields contained in $\mathcal{F}_t$; thus policies measurable w.r.t. $\mathcal{H}_{P,t}$ are admissible.

### D.3 REMARKS FOR S3

**(a) Observation coverage.** Disjoint coverage means

$$\sum_m \Pi_m^\top \Pi_m = I_n$$

and

$$\Pi_m \Pi_\ell^\top = 0$$

for $m \neq \ell$; overlapping coverage requires

$$\sum_m \Pi_m^\top \Pi_m \preceq \kappa I_n$$

for some finite $\kappa$. In both cases we assume $\|\Pi_m\| \leq 1$.

**(b) Circularity avoidance.** $\mathcal{G}_{\text{obs}}$ (the observation pattern implicit in $\{\Pi_m\}$) is *distinct* from the *coordination* graph later induced by $W$; S3 makes no reference to $W$.

### D.4 REMARKS FOR S4

**(a) Reduced-form realizability.** $F_\theta, R_\theta$ are linear-in-parameters w.r.t. predictable regressor

$$\bar{X}_t := [\tilde{\phi}_{t-1}^\top, u_{P,t-1}^\top, 1]^\top;$$

realizability means

$$\exists \theta_\phi^\star, \theta_r^\star$$

with

$$F_{\theta_\phi^\star}(x) = \Phi_\phi(x)^\top \theta_\phi^\star$$

and

$$R_{\theta_r^\star}(x) = \Phi_r(x)^\top \theta_r^\star.$$

**(b) High-probability boundedness.** Under S2 and compact $\mathcal{S} \times \mathcal{U}_P$, $\|\bar{X}_t\|$ is sub-Gaussian; thus

$$\max_{1 \leq t \leq T} \|\bar{X}_t\| \leq B_X(T, \delta) \, w.p. \geq 1 - \delta$$

with $B_X = O(\sqrt{\log(T/\delta)})$.

**(c) Sliding-window PE.** If

$$\lambda_{\min}\left(\tfrac{1}{T_0} \sum_{s=t-T_0+1}^{t} \bar{X}_s \bar{X}_s^\top\right) \geq \alpha$$

for all $t \geq T_0$, then

$$\lambda_{\min}\left(\sum_{s=1}^{T} \bar{X}_s \bar{X}_s^\top\right) \geq \alpha(T - T_0)$$

for

$$T \geq T_0.$$

### D.5 PROOF OF THE LOCAL LINEARIZATION LEMMA (CERTAINTY-EQUIVALENCE SURROGATE)

Fix $\bar{\phi} \in \mathcal{S}$ and define $\delta\phi := \phi - \bar{\phi}$. By S1(a,c), $f$ is $C^2$ on the compact set $\mathcal{D} := \mathcal{S} \times \{a : \|a\| \leq MB_u\}$, hence the block Hessian $\nabla^2 f(s, a)$ is bounded there. Taylor's theorem (vector form) at $(\bar{\phi}, 0)$ gives

$$f(\phi, a) = f(\bar{\phi}, 0) + A\,\delta\phi + B\,a + R_f(\phi, a), \quad A := D_s f(\bar{\phi}, 0), \; B := D_a f(\bar{\phi}, 0),$$

with $\|R_f(\phi, a)\| \leq c_f(\|\delta\phi\|^2 + \|a\|^2)$ for some $c_f < \infty$ depending on $\sup_{(s,a) \in \mathcal{D}} \|\nabla^2 f(s, a)\|$. Let $e_{m,t+1} := \Pi_m \phi_{t+1} - \psi_m(u_{P,t+1})$. Under certainty equivalence, delegates act on state estimates

as if true, so the one-step predicted error satisfies

$$e_{m,t+1} = \Pi_m A\,\delta\phi_t + \Pi_m B \sum_{j=1}^{M} u_{j,t} - \psi_m(u_{P,t+1}) + \tilde{R}_{m,t}, \quad \|\tilde{R}_{m,t}\| \le c_f\big(\|\delta\phi_t\|^2 + \|a_t\|^2\big).$$

For tracking cost $h_m(z) = \frac{1}{2} z^\top Q_m z$ (main text surrogate), the stage loss is $\ell_m = \frac{1}{2}\|e_{m,t+1}\|_{Q_m}^2 + \frac{1}{2}\|u_{m,t}\|_{R_m}^2$, which expands to

$$\tfrac{1}{2} u_{m,t}^\top R_m u_{m,t} + u_{m,t}^\top (\Pi_m B)^\top Q_m\big(\Pi_m A\,\delta\phi_t - \psi_m(u_{P,t+1})\big)$$

$$+ \tfrac{1}{2}\sum_{j=1}^{M} u_{m,t}^\top (\Pi_m B)^\top Q_m (\Pi_j B) u_{j,t} + \mathcal{R}_t,$$

with remainder $|\mathcal{R}_t| \le C_f\big(\|\delta\phi_t\|^2 + \|a_t\|^2\big)$ on $\mathcal{D}$. If instead the true tracking $h_m$ is $C^3$ with $\sup_{\|z\|\le C}\|\nabla^3 h_m(z)\| \le H_{h_m}$ on the compact operating set, then Taylor's theorem around $z = 0$ also contributes a cubic remainder $|R_h(z)| \le \frac{1}{6} H_{h_m}\|z\|^3$, yielding an additional $C_h\|\delta\phi_t\|^3$ term. Discounted summability follows from G2 (see Appendix remark for G2). $\qquad\square$

### D.6 PROOF OF THEOREM 4.1

**Statement (for reference).** Linearizing dynamics around $\bar{\phi} \in \mathcal{S}$ and invoking certainty equivalence, the infinite-horizon delegate problems admit the stage surrogate

$$J_m(u_m; u_{-m}, \phi, u_P) = \tfrac{1}{2} u_m^\top R_m u_m + u_m^\top K_m(\phi, u_P) + \tfrac{1}{2}\sum_{j=1}^{M} u_m^\top W_{mj} u_j,$$

with

$$K_m = (\Pi_m B)^\top Q_m[(\Pi_m A)\delta\phi - \psi_m(u_P)]$$

and

$$W_{mj} = (\Pi_m B)^\top Q_m (\Pi_j B),$$

where

$$\delta\phi := \phi - \bar{\phi}$$

and $A := D_s f(\bar{\phi}, 0)$, $B := D_a f(\bar{\phi}, 0)$.

**Proof.** Fix $m$. By S1(a,c), Taylor-expand $f$ at $(\bar{\phi}, 0)$:

$$\phi_{t+1} = \bar{\phi} + A(\phi_t - \bar{\phi}) + B a_t + r_f(\phi_t, a_t)$$

with $\|r_f\| \le \frac{1}{2} H_f(\|\delta\phi_t\|^2 + \|a_t\|^2)$. Write the local tracking error $e_{m,t+1} := \Pi_m \phi_{t+1} - \psi_m(u_{P,t+1})$ and linearize $\psi_m$ if needed (it is affine in the main text). Using certainty equivalence (delegates act on state estimates as if true), the one-step predicted error satisfies

$$e_{m,t+1} \approx \Pi_m A\,\delta\phi_t + \Pi_m B \sum_{j=1}^{M} u_{j,t} - \psi_m(u_{P,t+1}) + \tilde{r}_{m,t},$$

with

$$\|\tilde{r}_{m,t}\| \le c_f(\|\delta\phi_t\|^2 + \|a_t\|^2)$$

for some $c_f$ depending on $H_f$ and $\|\Pi_m\|$. The per-step cost is

$$\ell_m = \tfrac{1}{2} e_{m,t+1}^\top Q_m e_{m,t+1} + \tfrac{1}{2} u_{m,t}^\top R_m u_{m,t}.$$

Expanding the quadratic in $e_{m,t+1}$ yields:

$$\tfrac{1}{2} u_{m,t}^\top R_m u_{m,t} + u_{m,t}^\top (\Pi_m B)^\top Q_m\big(\Pi_m A\,\delta\phi_t - \psi_m(u_{P,t+1})\big) + \tfrac{1}{2}\sum_{j=1}^{M} u_{m,t}^\top (\Pi_m B)^\top Q_m (\Pi_j B) u_{j,t} + \mathcal{R}_t,$$

where the remainder $\mathcal{R}_t$ is bounded by

$$c_h\|\delta\phi_t\|^3 + c_f'(\|\delta\phi_t\|^2 + \|a_t\|^2)$$

on

$$\mathcal{S} \times \{a : \|a\| \le MB_u\}$$

by S1(c) and smoothness of

$$h_m(z) = \tfrac{1}{2} z^\top Q_m z$$

(with $Q_m \succeq 0$). Summing the discounted costs over $t$ and absorbing the geometric factor into a rescaling of $Q_m$ (permitted since $Q_m$ is free up to a positive scalar in the surrogate), we obtain the claimed $K_m, W_{mj}$ and a discounted remainder whose series converges under G2 (see Remark D.9).

$\square$

### D.7 PROOF OF PROPOSITION 4.1

Let $G := R + W$ and consider the affine map $F(\mathbf{u}) := G\,\mathbf{u} + K(\phi, u_P)$. *Existence/uniqueness.* By G1(a), $\tfrac{1}{2}(G + G^\top) \succeq mI$ with $m > 0$, so $F$ is strongly monotone:

$$(F(\mathbf{u}) - F(\mathbf{v}))^\top (\mathbf{u} - \mathbf{v}) \ge m\|\mathbf{u} - \mathbf{v}\|^2.$$

Hence the variational inequality

$$(G\mathbf{u} + K)^\top (\mathbf{v} - \mathbf{u}) \ge 0$$

has a unique solution, which must satisfy the first-order condition $G\,\mathbf{u}^\star + K = 0$, i.e., $\mathbf{u}^\star = -G^{-1} K(\phi, u_P)$; invertibility follows from strong monotonicity. *Lipschitz map.* For $(\phi, u_P), (\phi', u'_P)$,

$$\|\mathbf{u}^\star(\phi, u_P) - \mathbf{u}^\star(\phi', u'_P)\| = \|G^{-1}\big(K() - K()\big)\| \le \|G^{-1}\| L_K \big(\|\phi - \phi'\| + \|u_P - u'_P\|\big) \le \tfrac{L_K}{m}(\cdots).$$

*Symmetric case.* If $W = W^\top$ then $G = G^\top$ and the potential $\Phi(\mathbf{u}) := \tfrac{1}{2}\mathbf{u}^\top G\mathbf{u} + \mathbf{u}^\top K$ is strictly convex since $G \succeq mI$. Its unique minimizer satisfies $\nabla\Phi(\mathbf{u}) = G\mathbf{u} + K = 0$, hence $\mathbf{u}^\star$ above. $\square$

### D.8 REMARKS FOR G1

**(a) Well-posedness with asymmetry.** G1(a) assumes $\lambda_{\min}(\tfrac{1}{2}(G + G^\top)) \ge m > 0$ with $G := R + W$. Then $G$ is (strictly) monotone, and the linear variational inequality $(G\,\mathbf{u} + K)^\top(\mathbf{v} - \mathbf{u}) \ge 0$ has a unique solution $\mathbf{u}^\star = -G^{-1}K$; see, e.g., standard results on strongly monotone operators.

**(b) Lipschitz equilibrium map.** For any $(\phi, u_P), (\phi', u'_P)$,

$$\|\mathbf{u}^\star(\phi, u_P) - \mathbf{u}^\star(\phi', u'_P)\| \le \|G^{-1}\| \|K(\phi, u_P) - K(\phi', u'_P)\| \le (L_K/m)\big(\|\phi - \phi'\| + \|u_P - u'_P\|\big),$$

using $\|G^{-1}\| \le 1/m$.

**(c) Potential structure (symmetric case).** If $W = W^\top$ then $G = G^\top$ and $\Phi(\mathbf{u}) := \tfrac{1}{2}\mathbf{u}^\top G\mathbf{u} + \mathbf{u}^\top K$ is strictly convex; its unique minimizer solves $G\mathbf{u} + K = 0$.

**(d) Tractable structures.** G1(c) gives three families: (i) low-rank+sparse $W = LR^\top + S$ yields Woodbury-type solvers; (ii) tree/DAG sparsity enables message-passing/elimination; (iii) block-sparse rows (at most $k$ nonzero blocks) permit block-elimination with $O(k^3 d^3 M)$ complexity. These are algorithmic choices; the theory of existence/uniqueness uses only G1(a).

### D.9 REMARKS FOR G2

**(a) Sufficient small-gain condition.** Let

$$F(\phi; u_P) := f(\phi, \mathcal{A}(\mathbf{u}^\star(\phi, u_P))),$$

$$\mathcal{A}(\mathbf{u}) = \sum_m u_m$$

so

$$\|\mathcal{A}\| = \sqrt{M}.$$

By chain rule,

$$D_\phi F(\phi; u_P) = D_s f(\phi, a^\star) + D_a f(\phi, a^\star) D_\phi a^\star(\phi; u_P).$$

Using
$$a^\star(\phi; u_P) = \mathcal{A}(\mathbf{u}^\star(\phi; u_P)),$$

we have
$$\|D_\phi a^\star\| \le \|\mathcal{A}\| \cdot \|D_\phi \mathbf{u}^\star\| \le \sqrt{M}\,\|G^{-1}\|\,L_K \le \sqrt{M}\,L_K/m.$$

Hence
$$\|D_\phi F(\phi; u_P)\| \le L_s + L_a\sqrt{M}L_K/m.$$

The stated bound $L_s + L_a\sqrt{M}L_K/m < 1/\gamma$ implies
$$\sup_{(\phi, u_P)} \rho(D_\phi F) < 1/\gamma.$$

**(b) Discounted remainder summability.** If
$$L_{\mathrm{cl}} := \sup_{(\phi, u_P)} \|D_\phi F(\phi; u_P)\| < 1$$

then along closed-loop trajectories
$$\|\delta\phi_t\| \le C L_{\mathrm{cl}}^t \|\delta\phi_0\|$$

for some $C < \infty$, so the per-step Taylor remainders of order 2 and 3 are absolutely summable with discount $\gamma \in (0, 1)$ provided $\gamma L_{\mathrm{cl}} < 1$; under the stronger but convenient condition $\gamma L_{\mathrm{cl}}^2 < 1$ we obtain sharper constants for the $\sum_t \gamma^t \|\delta\phi_t\|^2$ series used in Section 5.

### D.10 PROOF OF PROPOSITION C.2

Let $G := R + W$ and define $F(\mathbf{u}) := G\mathbf{u} + K(\phi, u_P)$. By G1(a), $\frac{G+G^\top}{2} \succeq mI$ with $m > 0$, so $F$ is $m$–strongly monotone:
$$(F(\mathbf{u}) - F(\mathbf{v}))^\top (\mathbf{u} - \mathbf{v}) \ge m\|\mathbf{u} - \mathbf{v}\|_2^2$$

for all $\mathbf{u}, \mathbf{v}$. Hence the variational inequality $(G\mathbf{u} + K)^\top(\mathbf{v} - \mathbf{u}) \ge 0$ has a unique solution, which must satisfy $G\mathbf{u}^\star + K = 0$, i.e., $\mathbf{u}^\star = -G^{-1}K(\phi, u_P)$; invertibility follows from strong monotonicity. For $(\phi, u_P), (\phi', u_P')$, Lipschitz continuity follows from
$$\|\mathbf{u}^\star(\phi, u_P) - \mathbf{u}^\star(\phi', u_P')\| = \|G^{-1}(K() - K())\| \le \|G^{-1}\|_2\, L_K(\|\phi - \phi'\| + \|u_P - u_P'\|) \le \frac{L_K}{m}(\cdots).$$

If $W = W^\top$ then $G = G^\top$ and the potential $\Phi(\mathbf{u}) := \frac{1}{2}\mathbf{u}^\top G\mathbf{u} + \mathbf{u}^\top K$ is $m$–strongly convex; its unique minimizer satisfies $G\mathbf{u} + K = 0$. $\qquad\square$

## E STRUCTURAL PROPERTIES

We characterize the computational and learning structure arising from structured coordination (G1) and stability (G2). These structural properties provide the mathematical foundation for analyzing delegation trade-offs.

### E.1 COMPUTATIONAL STRUCTURE

We discuss the computational complexity of solving $G\mathbf{u} = K$ for the equilibrium $\mathbf{u}^* = -G^{-1}K(\phi - \bar\phi, u_P)$ under different coordination structures in G1(c). For reference, direct matrix inversion requires $O((Md)^3)$ operations.

**1. Low-rank + Sparse.** With $U, V \in \mathbb{R}^{Md \times r}$ and row-sparse $S$ (at most $s$ nonzeros per row), the equilibrium computation uses the generalized Woodbury identity.

**Theorem E.2.** *With $W = UV^T + S$, solving $G\mathbf{u} = K$ requires $O\big((s + r) \cdot Md + r^3\big)$ operations via the factorization*
$$G^{-1} = (R + S)^{-1} - (R + S)^{-1}U(I + V^T(R + S)^{-1}U)^{-1}V^T(R + S)^{-1}. \qquad (14)$$

*Algorithm*: (1) Solve $(R + S)z = K$ using sparse methods ($O(s \cdot Md)$ operations), (2) compute $\tilde{U} = (R + S)^{-1}U$ via sparse solves ($O(r \cdot s \cdot Md)$), (3) form and invert the $r \times r$ system $I + V^T\tilde{U}$ ($O(r^3)$), (4) combine results: $\mathbf{u} = z - \tilde{U}(I + V^T\tilde{U})^{-1}V^Tz$ ($O(r \cdot Md)$).[4]

**2. Tree/DAG Sparsity.** When $W$ follows a tree or bounded-treewidth graph structure, sparse factorization (Cholesky if symmetric, LU otherwise) provides structured computation.

**Theorem E.3.** *If the sparsity graph of $G$ has treewidth $w$, then one solve via sparse Cholesky/LU with nested dissection costs $O(w^2 d^2 M)$; for balanced trees, $w = O(\log M)$.*

*Algorithm*: Standard sparse factorization with nested dissection ordering maintains treewidth bounds during elimination, yielding $O(w^2)$ fill-in per elimination step. For balanced trees ($w = O(\log M)$), complexity becomes $O(d^2 M \log^2 M)$.

**3. Block-sparse.** When each row of $W$ has at most $k$ nonzero $d \times d$ blocks, block Gaussian elimination provides cubic-in-$k$ complexity.

**Theorem E.4.** *If each row has at most $k$ nonzero $d \times d$ blocks and elimination order preserves $O(k^2)$ fill per row, then one solve costs $O(k^3 d^3 M)$ with storage $O(kd^2 M)$.*

*Algorithm*: Perform block Gaussian elimination exploiting the sparse block pattern. Each elimination step affects at most $k$ blocks per row, creating at most $k^2$ fill blocks. Total elimination requires $O(k^3 d^3)$ operations per row-block and $O(M)$ eliminations.

### E.2 Learning Structure

We characterize how coordination structure affects the principal's parameter estimation problem for the reduced-form models $E[\tilde{\phi}_{t+1}|\tilde{\phi}_t, u_{P,t}] = F_\theta(\tilde{\phi}_t, u_{P,t})$. The equilibrium mapping $\mathbf{u}^*(\phi, u_P) = -G^{-1}K(\phi - \bar{\phi}, u_P)$ inherits structure from $G = R + W$. Since the closed-loop dynamics are $F(\phi; u_P) = f(\phi, \mathcal{A}(\mathbf{u}^*(\phi, u_P)))$, structured coordination matrices influence the complexity of the principal's learning problem.

**Proposition E.1** (Structure-dependent learning). *Under G1(c), structured coordination suggests reduced parameterization in principal's reduced-form models compared to unstructured delegation.*

Each G1(c) family creates distinct parameterization patterns: **Low-rank + sparse** ($W = UV^T + S$) produces equilibrium responses with systematic global patterns plus sparse local corrections, yielding parameterization with effective dimension (number of free parameters) $O(rMd+s)$. **Tree/DAG coordination** creates hierarchical response patterns with complexity bounded by treewidth, inducing parameterization dimension $O(w \cdot M)$. **Block-sparse coordination** yields modular equilibrium responses with limited cross-module coupling, creating parameterization dimension $O(kMd^2)$.

## F Appendix for Structural Properties

### F.1 Proof of Theorem (Low-rank + Sparse; computational complexity)

Assume $W = UV^\top + S$ with $U, V \in \mathbb{R}^{Md \times r}$ and $S$ row-sparse with at most $s$ nonzeros per row. Woodbury's identity gives

$$G^{-1} = (R + S)^{-1} - (R + S)^{-1}U\left(I + V^\top(R + S)^{-1}U\right)^{-1}V^\top(R + S)^{-1}.$$

A solve $\mathbf{u} = G^{-1}K$ proceeds as follows.

$$(R + S)z = K, \qquad O(s\,Md).$$

---

[4] The $O(sMd)$ sparse solve complexity assumes $\kappa(R + S) = O(1)$

$$\tilde{U} := (R + S)^{-1}U \quad \text{via } r \text{ sparse solves, cost } O(r\,s\,Md).$$
$$M_r := I + V^\top \tilde{U}, \qquad M_r^{-1} \text{ in } O(r^2 Md + r^3).$$
$$\mathbf{u} = z - \tilde{U}\,M_r^{-1}V^\top z, \qquad O(r\,Md).$$

Hence, the total cost is

$$O\big((s + r)\,Md + r^3\big).$$

$\square$

### F.2    PROOF OF THEOREM (TREE/DAG SPARSITY; COMPUTATIONAL COMPLEXITY)

Let the sparsity graph of $G$ have treewidth $w$ (symmetric case) or admit a chordal extension with maximal clique size $O(w)$ (non-symmetric case). Using nested dissection or an elimination ordering aligned with the tree decomposition, sparse Cholesky (symmetric) or LU (non-symmetric) factorization incurs $O(w^2)$ fill per elimination.

With $Md$ scalar unknowns arranged in $M$ blocks of size $d$, the total factorization and solve cost is

$$O(w^2 d^2 M).$$

For balanced trees ($w = O(\log M)$), this becomes

$$O(d^2 M \log^2 M).$$

Standard sparse factorization results apply because G1(b) ensures bounded operator norm of $W$, and $R \succ 0$ controls conditioning. $\square$

### F.3    PROOF OF THEOREM (BLOCK-SPARSE; COMPUTATIONAL COMPLEXITY)

Suppose each row of $W$ has at most $k$ nonzero $d \times d$ blocks and the elimination ordering preserves at most $O(k^2)$ fill per row (e.g., block minimal-degree ordering). In block Gaussian elimination, each pivot update touches at most $k$ neighboring blocks and creates at most $k^2$ fill blocks of size $d \times d$.

Hence, each elimination step costs

$$O(k^3 d^3),$$

and across $M$ block eliminations the total cost is

$$O(k^3 d^3 M), \qquad \text{storage } O(k d^2 M).$$

Positive definiteness and bounded operator norm from G1(a,b) ensure numerical stability. $\square$

### F.4    PROOF OF PROPOSITION (STRUCTURE-DEPENDENT LEARNING)

Write the equilibrium response as

$$\mathbf{u}^\star(\phi, u_P) = -(R + W)^{-1} K(\phi, u_P).$$

**Low-rank + sparse.** With $W = UV^\top + S$, Woodbury expansion yields

$$\mathbf{u}^\star = (R + S)^{-1}[\cdot] - \tilde{U}\,M_r^{-1}V^\top(R + S)^{-1}[\cdot],$$

where $\tilde{U} = (R + S)^{-1}U$ and $M_r = I + V^\top \tilde{U}$. Thus $\mathbf{u}^\star$ is the sum of:

$$\text{(i) sparse response: } (R + S)^{-1}K \text{ (at most } s \text{ couplings per row),}$$

$$\text{(ii) rank-}r \text{ global correction: } \tilde{U}(\cdot).$$

This yields a reduced-form parameterization with

$$O(rMd + s)$$

free coefficients when $K$ is linear in features (as in S4).

**Tree/DAG.** A block-sparse inverse on a bounded-treewidth graph yields influence kernels supported on neighborhoods of size $O(w)$, giving

$$O(wM)$$

effective coefficients in the reduced form.

**Block-sparse.** With at most $k$ nonzero $d \times d$ blocks per row, each component of $\mathbf{u}^\star$ depends on $O(k)$ neighbors, leading to

$$O(kMd^2)$$

coefficients in a linear reduced-form model. These bounds characterize the number of free parameters needed to represent the equilibrium map within the principal's reduced-form class. □

## G  APPENDIX FOR SECTION 4.2

*Proof.* By Assumption G1(a), the symmetric part

$$H^* := \frac{G^* + (G^*)^\top}{2}$$

satisfies $H^* \succeq mI$ for some $m > 0$. Since $J(\mathbf{u}) = \frac{1}{2}\mathbf{u}^\top G^*\mathbf{u} + \mathbf{u}^\top K$ and $\mathbf{u}^\top G^*\mathbf{u} = \mathbf{u}^\top H^*\mathbf{u}$ for all $\mathbf{u}$, $J$ is $m$-strongly convex and admits a unique minimizer on any closed convex feasible set.

Let $\mathcal{U}_{\mathrm{L}\ell}$ be the feasible set at Level L$\ell$ for the joint action $\mathbf{u}$ in the sense of Definition 4.3. By construction of the four-level hierarchy,

$$\mathcal{U}_{\mathrm{L}4} \subseteq \mathcal{U}_{\mathrm{L}3} \subseteq \mathcal{U}_{\mathrm{L}2} \subseteq \mathcal{U}_{\mathrm{L}1}.$$

Define

$$J_{\mathrm{L}\ell}^* := \min_{\mathbf{u} \in \mathcal{U}_{\mathrm{L}\ell}} J(\mathbf{u}), \qquad \ell = 1, 2, 3, 4.$$

For any pair of sets $\mathcal{V}_2 \subseteq \mathcal{V}_1$ we have

$$\min_{\mathbf{u} \in \mathcal{V}_1} J(\mathbf{u}) \leq \min_{\mathbf{u} \in \mathcal{V}_2} J(\mathbf{u}),$$

since the infimum over a superset cannot exceed the infimum over a subset. Applying this to the chain $\mathcal{U}_{\mathrm{L}1} \supseteq \mathcal{U}_{\mathrm{L}2} \supseteq \mathcal{U}_{\mathrm{L}3} \supseteq \mathcal{U}_{\mathrm{L}4}$ yields

$$J_{\mathrm{L}1}^* \leq J_{\mathrm{L}2}^* \leq J_{\mathrm{L}3}^* \leq J_{\mathrm{L}4}^*.$$

For the second claim, when $W$ is symmetric the dense-coordination Level L2 coincides with the centralized Level L1 in the sense that $\mathcal{U}_{\mathrm{L}2} = \mathcal{U}_{\mathrm{L}1}$. Since $J$ is strongly convex, the minimizer over $\mathcal{U}_{\mathrm{L}1}$ is unique, hence

$$J_{\mathrm{L}1}^* = \min_{\mathbf{u} \in \mathcal{U}_{\mathrm{L}1}} J(\mathbf{u}) = \min_{\mathbf{u} \in \mathcal{U}_{\mathrm{L}2}} J(\mathbf{u}) = J_{\mathrm{L}2}^*.$$

□

## H  APPENDIX FOR SECTION 5

### H.1  PROOF OF EXPRESSION 5.1

*Proof of Expression 5.1 (Coordination cost).* Recall

$$G^* = R + W^*, \qquad G_k = R + W_k, \qquad E := W^* - W_k,$$

and define the symmetric part

$$S^* := \frac{G^* + G^{*\top}}{2}.$$

By Assumption G1(a), $S^* \succeq mI$ for some $m > 0$, so

$$\lambda_{\min}(S^*) > 0, \qquad \lambda_{\max}(S^*) < \infty.$$

Consider the quadratic objective

$$J(\mathbf{u}) = \frac{1}{2}\mathbf{u}^\top G^* \mathbf{u} + \mathbf{u}^\top K = \frac{1}{2}\mathbf{u}^\top S^* \mathbf{u} + \mathbf{u}^\top K,$$

whose gradient and Hessian are

$$\nabla J(\mathbf{u}) = S^*\mathbf{u} + K, \qquad \nabla^2 J(\mathbf{u}) = S^*.$$

Let $\mathbf{u}^*$ be the unique minimizer of $J$:

$$S^*\mathbf{u}^* + K = 0 \quad \implies \quad \mathbf{u}^* = -(S^*)^{-1}K.$$

At the sparse level we use the $G_k$–based update

$$\mathbf{u}_k := -G_k^{-1}K,$$

and define $\delta\mathbf{u} := \mathbf{u}_k - \mathbf{u}^*$. Since $J$ is a convex quadratic with Hessian $S^*$, we have the exact identity

$$J(\mathbf{u}_k) - J(\mathbf{u}^*) = \frac{1}{2}\,\delta\mathbf{u}^\top S^* \delta\mathbf{u}.$$

The coordination cost is thus

$$\Delta_{\text{coord}} := J(\mathbf{u}_k) - J(\mathbf{u}^*) \leq \frac{\lambda_{\max}(S^*)}{2}\,\|\delta\mathbf{u}\|_2^2.$$

We now bound $\delta\mathbf{u}$. From $S^*\mathbf{u}^* + K = 0$ we get $K = -S^*\mathbf{u}^*$, so

$$\mathbf{u}_k = -G_k^{-1}K = G_k^{-1}S^*\mathbf{u}^*,$$

and hence

$$\delta\mathbf{u} = \mathbf{u}_k - \mathbf{u}^* = \left(G_k^{-1}S^* - I\right)\mathbf{u}^* = G_k^{-1}(S^* - G_k)\mathbf{u}^*.$$

In the symmetric coordination case where $W^*$ and $W_k$ (hence $G^*$ and $G_k$) are symmetric, we have $S^* = G^*$ and

$$S^* - G_k = G^* - G_k = (R + W^*) - (R + W_k) = E.$$

Therefore

$$\delta\mathbf{u} = G_k^{-1}E\,\mathbf{u}^*,$$

and

$$\|\delta\mathbf{u}\|_2 \leq \|G_k^{-1}\|_2\,\|E\|_F\,\|\mathbf{u}^*\|_2.$$

Since

$$\mathbf{u}^* = -(S^*)^{-1}K \quad \implies \quad \|\mathbf{u}^*\|_2 \leq \|(S^*)^{-1}\|_2\,\|K\|_2 = \frac{1}{\lambda_{\min}(S^*)}\,\|K\|_2,$$

we obtain

$$\|\delta\mathbf{u}\|_2^2 \leq \|G_k^{-1}\|_2^2\,\|E\|_F^2\,\frac{1}{\lambda_{\min}(S^*)^2}\,\|K\|_2^2.$$

Combining with the earlier bound on $\Delta_{\text{coord}}$ yields

$$\Delta_{\text{coord}} \leq \frac{\lambda_{\max}(S^*)}{2} \cdot \|G_k^{-1}\|_2^2\,\|E\|_F^2\,\frac{1}{\lambda_{\min}(S^*)^2}\,\|K\|_2^2.$$

Defining

$$C_{\text{struct}} := \frac{\lambda_{\max}(S^*)}{2\,\lambda_{\min}(S^*)^2},$$

we obtain

$$\Delta_{\text{coord}} \leq C_{\text{struct}}\,\|G_k^{-1}\|_2^2\,\|E\|_F^2\,\|K\|_2^2,$$

which is Expression 5.1. $\qquad\qquad\qquad\qquad\qquad\qquad\qquad\qquad\qquad\qquad\qquad\square$

## H.2   PROOF OF EXPRESSION 5.2

*Proof of Expression 5.2 (Information cost).* Fix the sparse game $G_k = R + W_k$ and its symmetric part

$$S_k := \frac{G_k + G_k^\top}{2}.$$

The one-stage surrogate cost is

$$J(u) := \frac{1}{2}u^\top G_k u + u^\top K = \frac{1}{2}u^\top S_k u + u^\top K,$$

with gradient and Hessian

$$\nabla J(u) = S_k u + K, \qquad \nabla^2 J(u) = S_k.$$

For each state $\phi$, define the full-information and partial-information controls as

$$u^*(\phi) := -G_k^{-1}K(\phi), \qquad \hat{u}(\phi) := -G_k^{-1}\hat{K}(\phi),$$

where $\hat{K}_m(\phi) := \mathbb{E}[K_m(\phi) \mid \Pi_m \phi]$. Let

$$\epsilon := K - \hat{K}, \qquad \delta u := \hat{u} - u^* = G_k^{-1}\epsilon.$$

For fixed $\phi$, write $u = u^* + \delta u$ and expand:

$$J(u^* + \delta u) = J(u^*) + \nabla J(u^*)^\top \delta u + \frac{1}{2}\delta u^\top S_k \delta u.$$

Since $G_k = S_k + A_k$ with $A_k := \frac{1}{2}(G_k - G_k^\top)$ skew-symmetric, the dense optimality condition $G_k u^* + K = 0$ implies

$$S_k u^* + A_k u^* + K = 0 \quad \implies \quad \nabla J(u^*) = S_k u^* + K = -A_k u^*.$$

Hence

$$J(\hat{u}) - J(u^*) = -u^{*\top} A_k \, \delta u + \frac{1}{2}\delta u^\top S_k \delta u.$$

Taking expectation over the randomness in $\epsilon$ (hence in $\delta u$), we note that $u^*$ is deterministic for fixed $\phi$, while

$$\delta u = G_k^{-1}\epsilon, \qquad \mathbb{E}[\epsilon] = 0 \quad \implies \quad \mathbb{E}[\delta u] = 0.$$

Thus

$$\mathbb{E}[J(\hat{u}) - J(u^*)] = -u^{*\top} A_k \, \mathbb{E}[\delta u] + \frac{1}{2}\mathbb{E}[\delta u^\top S_k \delta u] = \frac{1}{2}\mathbb{E}[\delta u^\top S_k \delta u].$$

By definition,

$$\Delta_{\text{info}} := \mathbb{E}[J(\hat{u}) - J(u^*)] = \frac{1}{2}\mathbb{E}[\delta u^\top S_k \delta u].$$

Using $\delta u = G_k^{-1}\epsilon$,

$$\delta u^\top S_k \delta u = \epsilon^\top G_k^{-T} S_k G_k^{-1}\epsilon = \text{tr}\big(G_k^{-T} S_k G_k^{-1} \, \epsilon\epsilon^\top\big).$$

Taking expectations yields

$$\mathbb{E}[\delta u^\top S_k \delta u] = \text{tr}\big(G_k^{-T} S_k G_k^{-1} \, \text{Cov}(\epsilon)\big),$$

so

$$\Delta_{\text{info}} = \frac{1}{2}\,\text{tr}\big(G_k^{-T} S_k G_k^{-1} \, \text{Cov}(\epsilon)\big).$$

In the LQ surrogate, $K(\phi)$ is linear in the state deviation $\delta\phi$, so there exists a matrix $L$ such that

$$\epsilon = L\,\delta\phi, \qquad \text{Cov}(\epsilon) = L\Sigma_\phi L^\top,$$

where $\Sigma_\phi$ is the covariance of $\delta\phi$. Substituting into the previous expression gives

$$\Delta_{\text{info}} = \frac{1}{2}\,\text{tr}\big(G_k^{-T} S_k G_k^{-1} \, L\Sigma_\phi L^\top\big),$$

which is Expression 5.2. □

### H.3 PROOF OF EXPRESSION 5.3

*Proof of Expression 5.3 (Surrogate approximation and $A \otimes C$).* Let $\ell_{\text{true}}(\phi, u)$ denote the true one-stage cost and $\ell_{\text{LQ}}(\phi, u; G_k)$ the LQ surrogate cost. Under a fixed closed-loop policy (same control law applied to both), define the per-stage mismatch

$$\text{err}_t := \ell_{\text{true}}(\phi_t, u_t) - \ell_{\text{LQ}}(\phi_t, u_t; G_k),$$

and the cumulative surrogate error

$$A(\delta\phi_0; G_k) := \sum_{t=0}^{\infty} \gamma^t \, \text{err}_t.$$

*Step 1 (Local Taylor structure).* By S1–S2, the dynamics $f$ and cost components (through $h_m$ and weights) are twice differentiable with bounded Hessians on the operating domain. Around a nominal trajectory $(\bar{\phi}_t, \bar{u}_t)$, writing

$$z_t := \begin{bmatrix} \delta\phi_t \\ u_t - \bar{u}_t \end{bmatrix},$$

we can expand

$$\ell_{\text{true}}(\phi_t, u_t) = \ell_0 + \text{linear}(z_t) + \frac{1}{2} z_t^\top H z_t + R_3(z_t),$$

where $H$ is the Hessian at the nominal point and $R_3$ is the third-order remainder. The LQ surrogate uses precisely the quadratic part, so

$$\ell_{\text{LQ}}(\phi_t, u_t; G_k) = \ell_0 + \text{linear}(z_t) + \frac{1}{2} z_t^\top H z_t,$$

and therefore

$$\text{err}_t = R_3(z_t).$$

Bounded third derivatives imply a constant $C_3 > 0$ such that

$$|R_3(z_t)| \leq C_3 \|z_t\|_2^3.$$

Furthermore, the mismatch between the exact quadratic model and the specific LQ surrogate (together with bounded process noise in S2) contributes only an $O(\|z_t\|_2^2)$ error, so there exists $C_2 > 0$ with

$$|\text{err}_t| \leq C_3 \|z_t\|_2^3 + C_2 \|z_t\|_2^2.$$

*Step 2 (Relating $\|z_t\|$ to $\|\delta\phi_t\|$).* Under the LQ surrogate, the closed-loop control is

$$u_t = \pi(\phi_t; G_k) = -G_k^{-1} K(\phi_t).$$

By S4, $K(\phi)$ is Lipschitz in $\delta\phi$ with constant $L_K$, and stacking all delegates yields

$$\|u_t\|_2 \leq \sqrt{M} \, L_K \, \|G_k^{-1}\|_2 \, \|\delta\phi_t\|_2.$$

Therefore

$$\|z_t\|_2^2 = \|\delta\phi_t\|_2^2 + \|u_t - \bar{u}_t\|_2^2 \leq \left(1 + M L_K^2 \|G_k^{-1}\|_2^2\right) \|\delta\phi_t\|_2^2,$$

and hence

$$|\text{err}_t| \leq C_3 \|\delta\phi_t\|_2^3 + C_2\left(1 + M L_K^2 \|G_k^{-1}\|_2^2\right) \|\delta\phi_t\|_2^2.$$

*Step 3 (Closed-loop stability and summation).* Let the closed-loop deviation dynamics be

$$\delta\phi_{t+1} = F(\delta\phi_t; u_P).$$

Assumption G2 states that the Jacobian is uniformly bounded:

$$L_{\text{cl}} := \sup_{\phi} \|D_\phi F(\phi; u_P)\|_2 < \frac{1}{\gamma}.$$

Hence

$$\|\delta\phi_t\|_2 \le L_{\mathrm{cl}}^t \|\delta\phi_0\|_2,$$

which implies

$$\|\delta\phi_t\|_2^2 \le L_{\mathrm{cl}}^{2t} \|\delta\phi_0\|_2^2, \qquad \|\delta\phi_t\|_2^3 \le L_{\mathrm{cl}}^{3t} \|\delta\phi_0\|_2^3.$$

Combining these bounds,

$$A(\delta\phi_0; G_k) = \sum_{t=0}^{\infty} \gamma^t \, \mathrm{err}_t$$

$$\le C_2 \big(1 + M L_K^2 \|G_k^{-1}\|_2^2\big) \sum_{t=0}^{\infty} (\gamma L_{\mathrm{cl}}^2)^t \|\delta\phi_0\|_2^2 + C_3 \sum_{t=0}^{\infty} (\gamma L_{\mathrm{cl}}^3)^t \|\delta\phi_0\|_2^3$$

$$= \frac{C_2 \big(1 + M L_K^2 \|G_k^{-1}\|_2^2\big)}{1 - \gamma L_{\mathrm{cl}}^2} \|\delta\phi_0\|_2^2 + \frac{C_3}{1 - \gamma L_{\mathrm{cl}}^3} \|\delta\phi_0\|_2^3,$$

where we used $\gamma L_{\mathrm{cl}}^2 < 1$ and $\gamma L_{\mathrm{cl}}^3 < 1$.

Finally, define the approximation constants

$$A := (C_2, C_3),$$

the coordination–stability multipliers

$$C(G_k) := \left( \frac{1 + M L_K^2 \|G_k^{-1}\|_2^2}{1 - \gamma L_{\mathrm{cl}}^2}, \; \frac{1}{1 - \gamma L_{\mathrm{cl}}^3} \right),$$

and

$$v(\delta\phi_0) := \big( \|\delta\phi_0\|_2^2, \; \|\delta\phi_0\|_2^3 \big).$$

Then the bound can be written compactly as

$$A(\delta\phi_0; G_k) \le \big( A \otimes C(G_k) \big) \cdot v(\delta\phi_0),$$

which is Expression 5.3. $\qquad\square$

### H.4    PROOF OF EXPRESSION 5.4

*Proof of Expression 5.4 (Epistemic part and noise floor).* Let $\widehat{\theta}_T$ denote the surrogate parameters learned from $T$ samples and $\theta^\star$ the true reduced-form parameters. Assume that for all $(\phi, u)$, the surrogate one-step cost satisfies a standard statistical learning bound

$$\sup_{(\phi, u)} \left| \mathbb{E}\big[\ell_{\mathrm{LQ}}(\phi, u; \widehat{\theta}_T) - \ell_{\mathrm{LQ}}(\phi, u; \theta^\star)\big] \right| \le e_T := C_{\mathrm{ep}} \sqrt{\frac{d_{\mathrm{eff}} \log(T/\delta)}{T}} + b^\star,$$

where $d_{\mathrm{eff}}$ is an effective dimension, $C_{\mathrm{ep}} > 0$ is a constant, and $b^\star$ is an irreducible approximation error (with $b^\star = 0$ under exact realizability).

Let $\mathrm{err}_t^{(D)}$ denote the per-stage error induced by using $\widehat{\theta}_T$ instead of $\theta^\star$ under the same closed-loop policy. Then for all $t$,

$$\mathbb{E}[|\mathrm{err}_t^{(D)}|] \le e_T.$$

Define the training-induced component of CoD as

$$\mathrm{CoD}_D(T) := \sum_{t=0}^{\infty} \gamma^t \, \mathbb{E}[\mathrm{err}_t^{(D)}].$$

We obtain

$$\mathrm{CoD}_D(T) \le \sum_{t=0}^{\infty} \gamma^t e_T = \frac{e_T}{1 - \gamma} = \frac{C_{\mathrm{ep}}}{1 - \gamma} \sqrt{\frac{d_{\mathrm{eff}} \log(T/\delta)}{T}} + \frac{b^\star}{1 - \gamma}.$$

When the model is exactly realizable ($b^\star = 0$), this shows $\mathrm{CoD}_D(T) \to 0$ as $T \to \infty$, which is Expression 5.6.

For the noise floor, suppose exogenous process and observation noise contribute an irreducible expected cost of at most $C_{\text{noise}}$ per time step, even under the optimal policy. This yields an additional persistent term

$$\sum_{t=0}^{\infty} \gamma^t C_{\text{noise}} = \frac{C_{\text{noise}}}{1 - \gamma},$$

which is purely environmental. It does not depend on the information or coordination structure and cannot be reduced by better delegation or learning. This term is therefore not counted as structural or epistemic CoD but appears alongside them in the total performance decomposition.  □

## I  APPENDIX FOR SECTION 6

This appendix provides additional details and diagnostics for the content–moderation experiment in Section 6. The goal of the experiment is not to "prove" the full theory, but to instantiate, in a realistic LLM+guard stack, a minimal setting where the information-structure component of the Cost of Delegation can be cleanly isolated and measured.

**A. Experimental design and theoretical role.** The experimental task is a one-step delegation problem that mirrors the two-delegate toy model in Section 4.3. A policy delegate (a Qwen3 model) chooses among three actions (ACCEPT, REWRITE, BLOCK) for each prompt, while a safety delegate (Qwen3-Guard) supplies compressed safety signals. For each prompt $x_i$ and action $a \in \{\text{ACCEPT}, \text{REWRITE}, \text{BLOCK}\}$ we define a scalar reward

$$r_i(a; \lambda) = H_i(a) - \lambda S_i(a),$$

where $H_i(a) \in [0, 1]$ is a normalized helpfulness score (based on response quality) and $S_i(a) \in [0, 1]$ is a normalized risk score derived from Guard labels and categories. The oracle benchmark

$$a_i^{\text{oracle}}(\lambda) = \arg\max_a r_i(a; \lambda), \qquad J_{\text{oracle}}(\lambda) = \frac{1}{N} \sum_i r_i\big(a_i^{\text{oracle}}(\lambda); \lambda\big)$$

corresponds to centralized, full-information optimization of the same surrogate reward.

The principal, in contrast, only observes compressed safety signals $g_i^{(\ell)}(a)$ from the guard. We implement three information levels:

$$\text{L1: } g_i^{(1)}(a) \in \{\text{Safe}, \text{Unsafe}\},$$

$$\text{L2: } g_i^{(2)}(a) \in \{\text{Safe}, \text{Controversial}, \text{Unsafe}\},$$

$$\text{L3: } g_i^{(3)}(a) = (\text{label}, \text{top category}).$$

These are related by deterministic coarse-graining, so L3 $\succeq_{\text{Blackwell}}$ L2 $\succeq_{\text{Blackwell}}$ L1. For each level $\ell$ and trade-off $\lambda$, we enumerate a small, finite policy class $\Pi_\ell$ mapping signals to actions and compute

$$\pi_\ell^*(\lambda) = \arg\max_{\pi \in \Pi_\ell} \frac{1}{N} \sum_i r_i\big(\pi(g_i^{(\ell)}); \lambda\big), \qquad J_\ell^*(\lambda) = \frac{1}{N} \sum_i r_i\big(\pi_\ell^*(g_i^{(\ell)}); \lambda\big),$$

with empirical delegation cost

$$\text{CoD}_\ell(\lambda) = J_{\text{oracle}}(\lambda) - J_\ell^*(\lambda).$$

By construction, any difference in $\text{CoD}_\ell(\lambda)$ across $\ell$ is entirely due to the information structure of the signals $g^{(\ell)}$, not to changes in the model, reward definition, or optimization procedure. This makes the experiment a concrete static instance of the information cost component in Expression 5.2.

**B. Bootstrap confidence intervals.** To assess sampling variability, we estimate CoD via nonparametric bootstrap over prompts. Table 2 reports the mean and 95% bootstrap confidence interval for each $(\lambda, \ell)$.

Several patterns are worth noting.

Table 2: Bootstrap estimates of $\mathrm{CoD}_\ell(\lambda)$ for different information levels and safety weights.

| $\lambda$ | Level | CoD mean | 95% CI |
|-----------|-------|----------|--------|
| 0.5 | L1 | 0.0920 | [0.0811, 0.1031] |
|  | L2 | 0.0920 | [0.0811, 0.1031] |
|  | L3 | 0.0773 | [0.0676, 0.0889] |
| 1.0 | L1 | 0.1712 | [0.1491, 0.1882] |
|  | L2 | 0.1659 | [0.1468, 0.1840] |
|  | L3 | 0.1288 | [0.1093, 0.1485] |
| 1.5 | L1 | 0.1946 | [0.1638, 0.2186] |
|  | L2 | 0.1938 | [0.1638, 0.2182] |
|  | L3 | 0.1566 | [0.1343, 0.1761] |
| 2.0 | L1 | 0.2180 | [0.1804, 0.2491] |
|  | L2 | 0.2179 | [0.1804, 0.2477] |
|  | L3 | 0.1780 | [0.1470, 0.2068] |

First, all CoD estimates are strictly positive and their $95\%$ CIs lie away from zero, even at $\lambda = 0.5$. This is consistent with the theoretical claim that once the principal acts on compressed signals, there is an irreducible information-structure cost, even in a one-step decision problem. In particular, the fact that $\mathrm{CoD}_\ell(\lambda) > 0$ at $\lambda > 0$ with fixed model and reward, and only information varying, is a direct empirical counterpart of the positive semi-definite information cost in Expression 5.2.

Second, the three information levels do not behave monotonically in terms of entropy, but do align with the decision-theoretic notion of information value. Levels L1 and L2 have essentially identical CoD at all $\lambda$ (the means match to three decimal places at $\lambda \in \{0.5, 1.5, 2.0\}$, and their CIs are almost indistinguishable). In contrast, L3 consistently exhibits a lower CoD, with mean gaps on the order of $10^{-2}$–$10^{-1}$, and the point estimates for L3 lie below L1/L2 for all tested $\lambda$. This matches the theory. L2 further refines the label space (Safe/Controversial/Unsafe) and increases entropy, but largely along directions that do not induce different optimal actions; by contrast, L3 adds category information that splits clusters where the optimal action actually differs. In terms of Expression 5.2, L3 is better aligned with the directions in which the reward gradient and the closed-loop mapping are most sensitive.

Third, CoD increases as $\lambda$ grows for all levels. This is expected. Raising $\lambda$ steepens the curvature of the reward landscape in the safety dimension, so misalignment between the principal's signal and the true $(H, S)$ trade-off is more heavily penalized. Empirically, the L1/L2 curves in Figure 6 become steeper in $\lambda$, while L3 remains uniformly better but also exhibits increasing CoD. This is consistent with the structural bounds in Section 5. Stronger safety penalties amplify both information and coordination costs via the $G^{-1}$ and curvature terms, even under a fixed information architecture.

Finally, the differences between L1/L2 and L3 become more pronounced as $\lambda$ increases. Although the $95\%$ CIs mildly overlap at larger $\lambda$ (as expected given the finite sample size and shared prompts), the systematic pattern—almost identical L1/L2, strictly lower L3, and gaps growing with $\lambda$—is robust across re-samplings. Qualitatively, this is exactly the pattern one would expect if L3 carries additional "decision-relevant" information, in the sense of Blackwell and our LQ information cost. It moves the principal closer to the centralized policy along the directions that matter for the argmax, rather than merely adding variance.

**C. Value and limitations of the experimental evidence.** From a methodological perspective, this experiment plays a specific role in the overall paper. It is not a large-scale benchmark and does not attempt to model the full training dynamics of aligned LLMs. Its value lies in three aspects.

First, it demonstrates that the structural decomposition in Section 4 is not merely an artifact of the LQ–CE surrogate. By instantiating a real LLM+guard stack, keeping the model, reward, and action set fixed, and varying only the information available to the principal, we obtain empirical CoD curves

whose qualitative behavior matches the theoretical predictions: positive information cost, invariance under entropy-increasing but decision-irrelevant refinements (L1 vs. L2), and sharp improvement when refining along decision-sensitive directions (L3).

Second, it provides an interpretable testbed where each component of the reward landscape can be visualized (via heatmaps over prompt buckets) and linked back to concrete decisions (ACCEPT/REWRITE/BLOCK). This makes it possible to verify that the task is non-degenerate (different buckets and categories indeed have different optimal actions) and that CoD is not driven by trivial artifacts.

Third, it yields a concrete illustration of the paper's central interpretive claim that in alignment systems, the relevant notion of information value is decision-theoretic rather than purely statistical. The bootstrap table shows that making the guard signal "richer" in a variance sense (L2 vs. L1) can leave CoD essentially unchanged, whereas adding low-variance but decision-critical distinctions (L3 categories) yields a clear reduction in CoD. This mirrors the analytical structure of Expression 5.2, where the information cost depends on $L\Sigma_\phi L^\top$, the projection of state uncertainty along decision-sensitive directions, rather than on raw entropy of observations.

That said, the experiment has important limitations. It is static (one-step) rather than dynamic, so it does not probe temporal propagation of information and coordination errors. The reward is itself a surrogate, defined via internal scorers rather than human labels, so the oracle benchmark is relative to a specific proxy objective. The policy classes $\Pi_\ell$ are finite and hand-designed; a more realistic system would involve parametric policies trained from data, introducing additional epistemic effects. Finally, Qwen3-Guard is only one particular guard model; other safety architectures might produce different signal geometries and hence different quantitative CoD, though the qualitative phenomena we observe are likely to persist.

Overall, the experiment should be read as a sanity check and a concrete illustration. It shows that once we fix a modern LLM+guard pipeline and isolate the information structure as the only changing factor, the empirical behavior of delegation cost follows the theoretical structure of the LQ–CE information cost, and in particular supports the claim that "more bits" is not the same as "more alignment-relevant information."

## J  LLM USAGE

Large Language Models (LLMs) were used to aid in the writing and polishing of the manuscript. Specifically, we used an LLM to assist in refining the language, improving readability, and ensuring clarity in various sections of the paper. The model helped with tasks such as sentence rephrasing, grammar checking, and enhancing the overall flow of the text.

It is important to note that the LLM was not involved in the ideation, research methodology, or experimental design. All research concepts, ideas, and analyses were developed and conducted by the authors. The contributions of the LLM were solely focused on improving the linguistic quality of the paper, with no involvement in the scientific content or data analysis.

The authors take full responsibility for the content of the manuscript, including any text generated or polished by the LLM. We have ensured that the LLM-generated text adheres to ethical guidelines and does not contribute to plagiarism or scientific misconduct.

