# OpenReview forum: "The Cost of Delegation"
_ICLR.cc/2026/Conference — ICLR 2026 Conference Withdrawn Submission_

### Official Review · Reviewer_beuH · 2025-10-25

**Soundness:** 2
**Presentation:** 3
**Contribution:** 2
**Rating:** 4
**Confidence:** 2

**Summary:**

The paper studies hierarchical delegation in multi-agent control systems. It formalizes how a principal can guide multiple delegates under partial observability, analyzes the resulting Nash equilibrium, and quantifies the approximation error (termed “cost of delegation”) arising from quadratic surrogates of the full nonlinear dynamics. The framework distinguishes between epistemic (learning) errors and persistent structural errors, providing bounds and design principles to reduce the latter.

**Strengths:**

Rigorous formal treatment connecting multi-agent delegation, surrogate LQ games, and closed-loop stability.

Clear separation of epistemic versus persistent errors.

Provides practical guidance for designing hierarchical control systems.

**Weaknesses:**

The title “cost of delegation” is potentially misleading, conflating LQ surrogate approximation error with intrinsic delegation inefficiency.

Heavy notation and long derivations reduce accessibility.

Lacks numerical or empirical examples to illustrate the bounds in practice.

**Questions:**

Can you explicitly explain the relationship between LQ approximation error and delegation inefficiency, e.g., due to partial observability or divergence between delegate and principal objectives? Are you assuming that under quadratic approximation the delegation inefficiency vanishes (even if the delegates are misaligned or adversarial), and thus the divergence from LQ approximation can indeed bound the cost of delegation?

---

### Official Review · Reviewer_Jvyk · 2025-11-01

**Soundness:** 2
**Presentation:** 3
**Contribution:** 3
**Rating:** 6
**Confidence:** 2

**Summary:**

This paper provides a theoretical framework to explain when delegation enhances or harms organizational performance, highlighting its dependence on information asymmetry and preference alignment. It models the trade-off between improved information acquisition by agents (when decisions are delegated) and the loss of centralized control by principals. The authors find that delegation can be costly when agents’ preferences diverge from the principal’s, leading to inefficiencies, but beneficial when it motivates better information gathering or reduces communication frictions.

**Strengths:**

1. The paper offers a clean, well-structured model that clearly formalizes the trade-off between control and information in delegation decisions.
2. It advances understanding of when delegation improves or harms efficiency, providing a foundation for later empirical and behavioral studies.

**Weaknesses:**

1. The model relies on strong assumptions of linear assumptions, full information, and simple preference structures.
2. The paper does not test its theoretical predictions with data or case studies, limiting practical relevance.
3. The paper has a discussion of its connection with modern systems like RLHF, multi-agent systems, and MoE training, but does not disclose detailed guidance on that.

**Questions:**

1. Can the authors raise some examples on how the cost of delegation affect current modern systems?
2. How might the main results change if the system dynamics were highly nonlinear or involved deep-learning systems?

---

### Official Review · Reviewer_TKto · 2025-11-03

**Soundness:** 3
**Presentation:** 2
**Contribution:** 3
**Rating:** 6
**Confidence:** 2

**Summary:**

The paper views multi-agent reinforcement learning and alignment through the lens of hierarchical coordination, where a principal directs multiple partially informed and interdependent delegates. Building on first principles with certainty-equivalence and local linearization, it develops a linear–quadratic surrogate in which the delegates’ interactions form a quadratic game whose Nash equilibrium determines overall behavior, yielding an explicit equilibrium map. The authors study three coordination structures (low-rank plus sparse, tree/DAG, and block-sparse), showing how each yields tractable solvers and predictable computational scaling. Finally, it formalizes the Cost of Delegation (CoD) as a persistent, non-vanishing penalty that decomposes into a cubic value-function remainder and a quadratic dynamics remainder, which distinguishes it from epistemic error.

**Strengths:**

1. The paper provides a clear framing of alignment as hierarchical coordination between a principal and many coupled delegates.
2. The paper give clean first-principles derivation of an LQ surrogate and explicit equilibrium map with uniqueness under standard conditions.
3. The paper novelly introduces the cost of delegation, which is a persistent penalty decomposed into a cubic value-function remainder and a quadratic dynamics remainder, distinct from epistemic error.

**Weaknesses:**

1. While I find the results very interesting, the writing is not strong. At present, the paper reads like a list of theorems, definitions, and assumptions. Adding more discussion and explanatory text would make it much more readable. The paper also needs a more detailed introduction and a more comprehensive discussion of related work.

2. As noted in the limitations paragraph, the analyzed games are quadratic surrogate games, which are far removed from modern AI systems.

3. See questions.

**Questions:**

1. In Theorem 3.1, shouldn’t it be $W_{mj} = (\Pi_m B)^\top Q_m (\Pi_m B)$ instead?

2. In line 360, the paper states that the effective dimension is (O(rMd + s)), whereas line 729 in the appendix gives (O(rMd + sM)). Could the authors clarify this?

---

### Official Review · Reviewer_8hwx · 2025-11-11

**Soundness:** 3
**Presentation:** 3
**Contribution:** 3
**Rating:** 6
**Confidence:** 1

**Summary:**

This paper presents a framework for understanding multi-agent reinforcement learning and alignment through hierarchical coordination, where a principal steers multiple delegates with partial observability. The core contribution is formalizing and quantifying the Cost of Delegation - an irreducible performance penalty inherent to indirect control.

**Strengths:**

1. Applicability - Despite being highly theoretical, the paper connects well to practical systems (RLHF, Constitutional AI, MoE architectures). The design principles in Section 5.2 offer actionable insights for system designer. I feel that there is likely some nuance in translating the theoretical nuance of this paper to modern systems, but nevertheless, its theoretical framing of hierarchical delegation costs is interesting and novel.
2. Theory - The analysis of computational complexity under different coordination structures (low-rank+sparse, tree/DAG, block-sparse) is thorough and provides practical guidance for scalable implementations.

**Weaknesses:**

1. No Empirical Investigation - As an ICLR submission, I would assume the paper has some experiments that demonstrate practical applicability. I feel the paper could be strengthened with some application to MoEs.

**Questions:**

In a scenario where we have several experts over several layers, how does the cost of delegation scale?
- are there ways of bounding cross-entropy loss according to the CoD framework?

---

### Author Response · Authors · 2025-11-28
**Update on Our Revisions**

Dear Reviewers,

Thank you for your patience during the discussion phase. We want to provide a concise update on the status of our revision process.

We are currently completing a substantial overhaul of the original manuscript. The revised version introduces fundamentally new insights that directly address the core concerns raised in the initial reviews and significantly strengthen the technical soundness of our framework. We have also restructured the exposition, adding clarifying discussion, explanatory text, and illustrative figures to improve readability and interpretability. In addition, we have also added an experiment to validate the claims.

We apologize for the delay in submitting our responses. We will upload the updated manuscript shortly and provide a detailed, point-by-point reply to each comment.

Thank you again for the constructive feedback and for your understanding during the discussion period!

Best regards,
Authors of Submission373

---

### Note · Authors · 2026-01-26

I have read and agree with the venue's withdrawal policy on behalf of myself and my co-authors.

---

### Meta-Review · Area_Chair_Tj9C · 2026-01-04

**Summary:**

The paper proposes a theoretical framework to quantify the "Cost of Delegation" (CoD) in hierarchical multi-agent systems using a Linear-Quadratic (LQ) surrogate. The decision to reject, despite the surface-level positive scores (6, 6, 6, 4), is driven by three critical concerns shared by the reviewers and the AC regarding the paper's relevance and impact:
- The Theory-Practice Gap: As highlighted by Reviewers TKto and Jvyk, the reliance on LQ surrogates and strong stability assumptions creates a fundamental disconnect from the complex, non-linear dynamics of modern deep learning systems (e.g., LLMs). This makes the practical implications of the theory largely speculative.
- Limited Actionable Insight: The core finding—that systems should prioritize decision-relevant features—is effectively a formalization of the classical "Value of Information" within a specific control framework. It offers limited new, actionable guidance for modern system design beyond high-level heuristics, failing to justify the complexity for the general ML community (Reviewer beuH).
- Accessibility Barrier: The heavy reliance on dense control theory formalism creates a high barrier to entry. As noted by Reviewers beuH and TKto, this makes the paper inaccessible to a large portion of the ICLR audience, significantly limiting its potential impact.

These outstanding concerns explain the extremely low reviewer confidence (1, 2, 2, 2) and substantiate the decision that the paper is technically correct but not suitable for ICLR in its current form.

**Reviewer Concerns:**

Addressed by Rebuttal:
- Lack of Experiments: The authors added a static content moderation experiment (Qwen + Guard) to illustrate the theoretical bounds. This addressed the literal "no experiment" complaint from Reviewer 8hwx and Jvyk.
- Structure and Clarity: The authors restructured the paper and added narrative explanations, which partially alleviated Reviewer TKto's concern about the paper reading like a "list of theorems."

Outstanding Concerns (Basis for Rejection):
- The Gap between Theory and Practice: This is the critical outstanding issue. As noted by Reviewers TKto and Jvyk, and reinforced by the AC's assessment, the reliance on LQ surrogates and strong stability assumptions (G2) creates a significant gap when applying these insights to modern, highly non-linear deep learning systems (like LLMs). The leap from LQG control theory to RLHF/MoE dynamics is substantial and speculative.
- Accessibility and Impact: As explicitly noted by Reviewers beuH and TKto, the heavy reliance on control theory notation makes the paper read like a "list of theorems," reducing its accessibility to the broader ICLR audience. This barrier, combined with the lack of direct actionable guidance for non-linear systems, limits the paper's potential impact.
- Limited Contribution: Despite the mathematical rigor, reviewers rated the contribution as only "fair" (beuH) or questioned the practical relevance (Jvyk), indicating that the theoretical insights—while self-consistent—do not justify the complexity for the general machine learning community.

**Reviewer Scores:**

The current scores (6, 6, 6) appear inflated due to the "survival bias" of theoretical papers where reviewers cannot find technical flaws but fail to appreciate the significance.
- Reviewer 8hwx (Score 6 -> Withdraw): Explicitly admitted to lacking background and effectively withdrew. Their score should be disregarded.
- Reviewer TKto & Jvyk (Score 6 -> 4/5): If they had fully engaged with the limitation that the LQ assumptions may not transfer to deep learning dynamics (making the practical implications speculative), their scores would likely drop to "Weak Reject" or "Borderline Reject."
- Reviewer beuH (Score 4 -> 4): Would likely maintain the Reject score as the fundamental concern about the disconnect between the "Cost of Delegation" framing and the LQ approximation error remains valid.

---

### Decision · Program_Chairs · 2026-01-26

Reject